

# Speciation among sympatric lineages in the genus *Palythoa* (Cnidaria: Anthozoa: Zoantharia) revealed by morphological comparison, phylogenetic analyses and investigation of spawning period

Masaru Mizuyama[1], Giovanni D. Masucci[1] and James D. Reimer[2,3]

[1] Molecular Invertebrate Systematics and Ecology Laboratory, Graduate School of Marine Science, University of the Ryukyus, Nishihara, Okinawa, Japan
[2] Molecular Invertebrate Systematics and Ecology Laboratory, Department of Marine Sciences, Chemistry and Biology, Faculty of Science, University of the Ryukyus, Nishihara, Okinawa, Japan
[3] Tropical Biosphere Research Center, University of the Ryukyus, Nishihara, Okinawa, Japan

Corresponding author
James D. Reimer,
jreimer@sci.u-ryukyu.ac.jp

## ABSTRACT

Zoantharians are sessile marine invertebrates and colonial organisms possessing sexual and asexual reproductive ability. The zooxanthellate zoantharian genus *Palythoa* is widely distributed in coral reef ecosystems. In the Ryukyu Archipelago, Japan, sympatric *Palythoa tuberculosa* and *P. mutuki* are the dominant species of this genus in the intertidal zone. Previous phylogenetic analyses have shown that these two species are closely related, and additionally revealed a putative sympatric hybrid species (designated as *Palythoa* sp. yoron). In this study, we attempted to delineate *Palythoa* species boundaries and to clarify the relationships among these three groups plus another additional putative sympatric species (*P.* aff. *mutuki*) by multiple independent criteria. The morphology of these four lineages was clearly different; for example the number of tentacles was significantly different for each species group in all pairwise comparisons. From observations of gonadal development conducted in 2010 and 2011, *P.* sp. yoron and *P.* aff. *mutuki* appear to be reproductively isolated from *P. tuberculosa*. In the phylogenetic tree resulting from maximum likelihood analyses of the ITS-rDNA sequence alignment, *P. tuberculosa* and *P.* sp. yoron formed a very well supported monophyletic clade (NJ = 100%, ML = 95%, Bayes = 0.99). This study demonstrates that despite clear morphological and/or reproductive differences, *P. tuberculosa* and *P.* sp. yoron are phylogenetically entangled and closely related to each other, as are *P. mutuki* and *P.* aff. *mutuki*. Additionally, no single molecular marker was able to divide these four lineages into monophyletic clades by themselves, and a marker that has enough resolution to solve this molecular phylogenetic species complex is required. In summary, the morphological and reproductive results suggest these lineages are four separate species, and that incomplete genetic lineage sorting may prevent the accurate phylogenetic detection of distinct species with the DNA markers utilized in this study, demonstrating the value of morphological and reproductive data when examining closely related lineages.

# INTRODUCTION

Zoantharians are sessile marine invertebrates and colonial organisms possessing sexual and asexual reproductive ability (*Ryland, 1997*). Zoantharians belong to subclass Hexacorallia (Cnidaria, Anthozoa) and they have the significant feature of embedding small particles (sand, detritus) into their body column. Zooxanthellate zoantharian species are found worldwide in tropical and subtropical shallow water areas (*Trench, 1974*; *Reimer, Takishita & Maruyama, 2006*).

Traditionally, zoantharian classification has been based on morphological characters such as the relative degree of coenenchyme development, number of tentacles per polyp, oral disk diameter, and position and features of the sphincter muscle (*Ryland & Lancaster, 2003*). However, sand encrustation (*Reimer et al., 2010*) and large intraspecific variation have often made histological classification difficult (*Muirhead & Ryland, 1985*; *Mueller & Haywick, 1995*; *Reimer et al., 2010*). Phylogenetic work using mitochondrial 16S ribosomal DNA and cytochrome oxidase subunit I (mtCOI) and the nuclear internal transcribed spacer region of ribosomal DNA (ITS-rDNA) as molecular markers have begun to reveal evolutionary relationships in this group (e.g., *Reimer et al., 2004*; *Sinniger et al., 2005*; *Reimer et al., 2007b*).

The zooxanthellate zoantharian genus *Palythoa* Lamouroux, 1816 is widely distributed in coral reef ecosystems as a common group of organisms. In the Ryukyu Archipelago of southern Japan (Fig. 1), *Palythoa tuberculosa* (Esper, 1805) and *P. mutuki* Haddon & Shackleton, 1891 are the dominant species of this genus in the intertidal zone (*Irei, Nozawa & Reimer, 2011*). *Reimer et al. (2007a)* showed that these two species are closely related with phylogenetic analyses based on ITS-rDNA and mtCOI. Furthermore, they revealed a putative hybrid species (designated as *Palythoa* sp. yoron), which was presumed to have originated via interspecies hybridization between *P. tuberculosa* and *P. mutuki*, based on shared additive patterns of nucleotide polymorphisms of ITS-rDNA sequences, and indicated a potential reticulate evolutionary history in these three species groups. A subsequent investigation conducted by *Shiroma & Reimer (2010)* revealed that *P.* sp. yoron was sympatric in the intertidal zone with these two other species in Okinawa, but also was present in a different microenvironment than *P. tuberculosa* and *P. mutuki*. As well, *P.* sp. yoron is intermediate in morphological form between *P. tuberculosa* and *P. mutuki*. (Fig. 2, Table 1), with all three species readily distinguishable from one another (*Shiroma & Reimer, 2010*).

In this study, we attempted to determine the delimitation of *Palythoa* species boundaries and to clarify the relationships among species groups using multiple independent criteria. We first made primary hypotheses of species delimitation based on morphology and habitat preference. We then re-examined these hypotheses via genetic data and investigated ovary
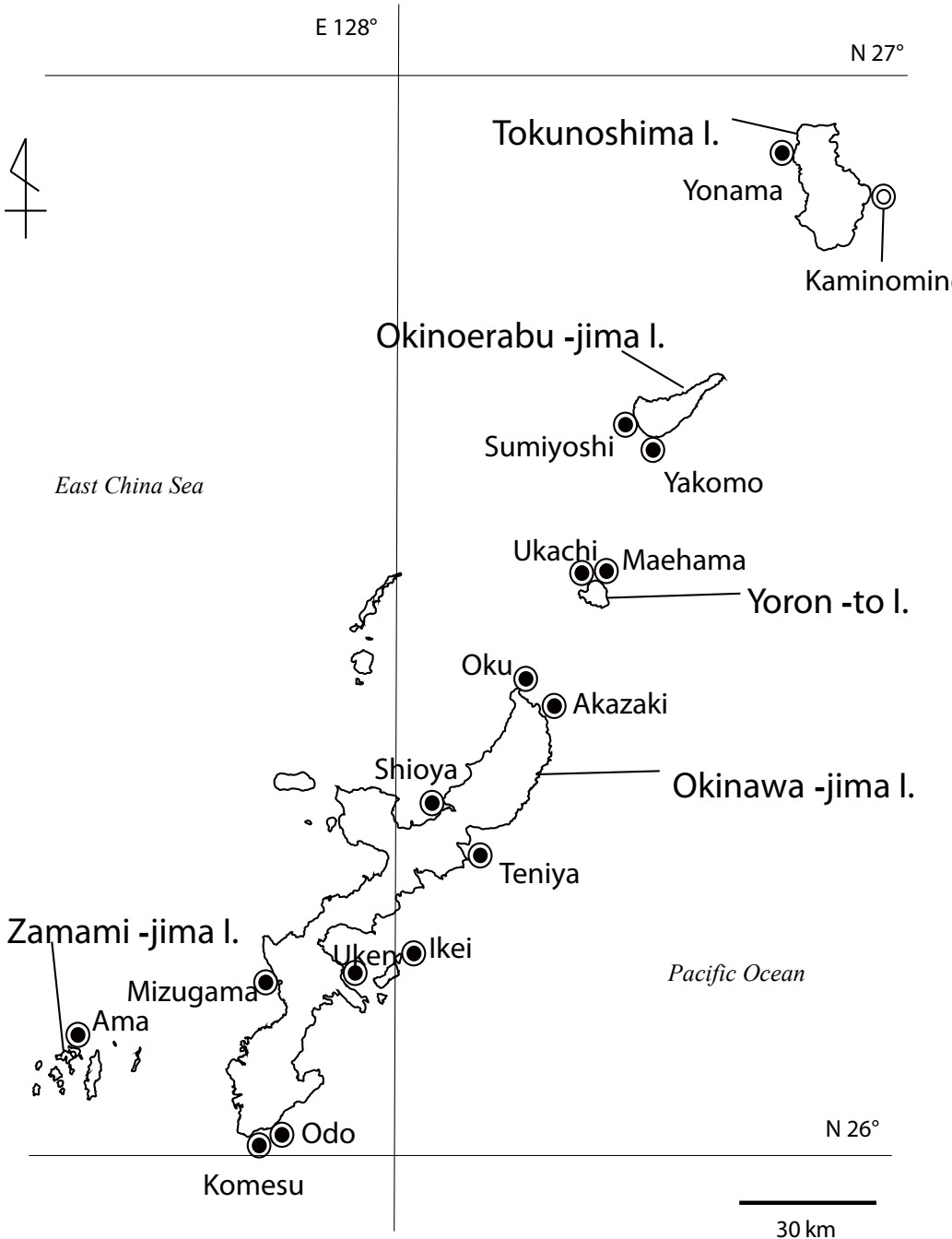

**Figure 1** **Map of Palythoa species specimen locations in the Ryukyu Archipelago in this study.** Map of *Palythoa* species specimen locations in the Ryukyu Archipelago, including Okinawa-jima I., Zamami-jima I., Yoron-to I., Okinoerabu-jima I., and Tokunoshima I. Locations for specimens collected in this study represented by closed symbols, location for spawning timing investigations represented by open symbol.

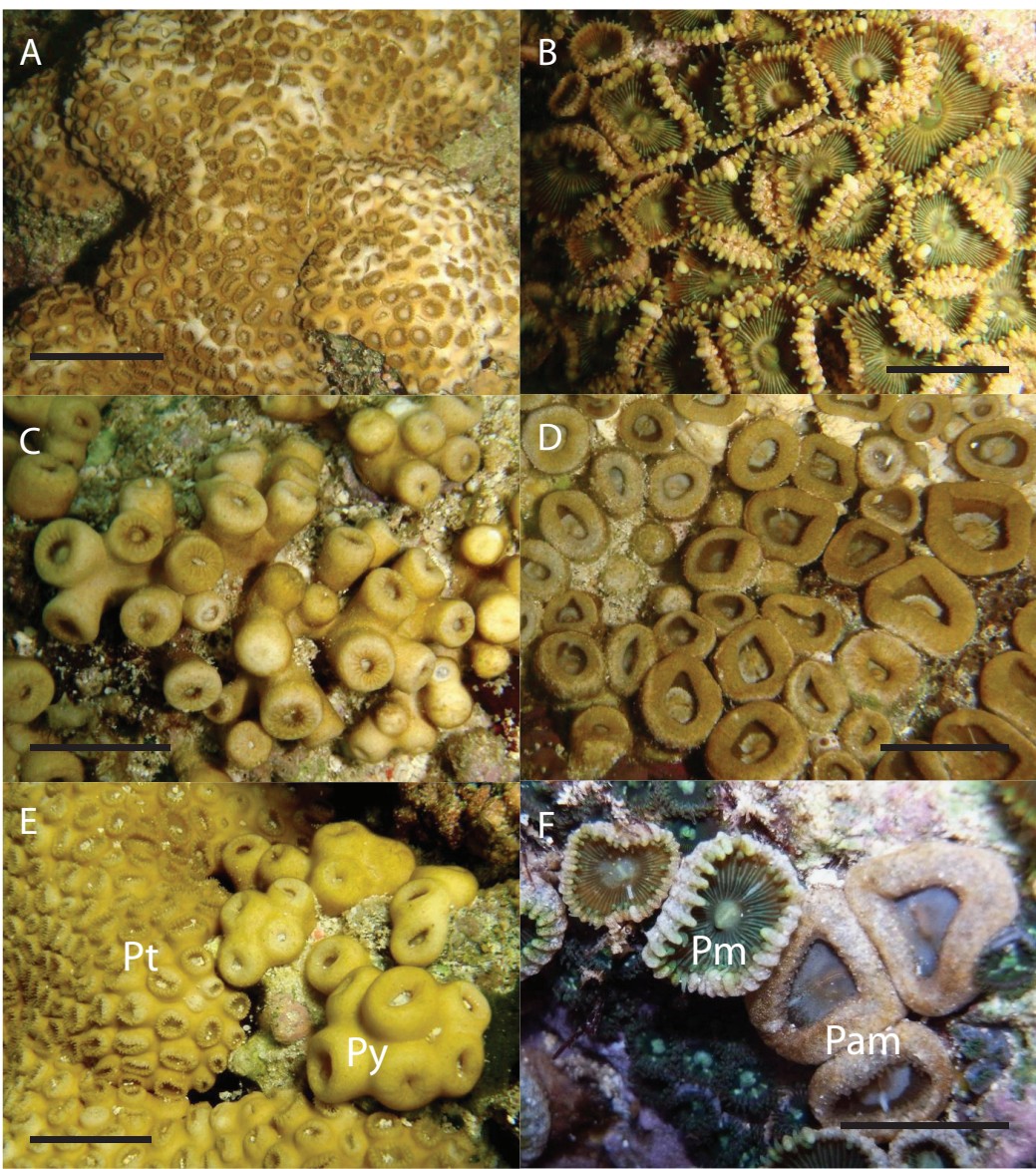

**Figure 2** ***In situ* images of *Palythoa* species examined in this study.** *In situ* images of (A) *Palythoa tuberculosa*, (B) *P. mutuki*, (C) *P.* sp. yoron, (D) *P.* aff. *mutuki*, (E) *P. tuberculosa* (left; "Pt") and *P.* sp. yoron (right, "Py"), and (F) *P. mutuki* (left, "Pm") and *P.* aff. *mutuki* (right, "Pam"). Scale bars in (A), (C), (E) are 2 cm, in (B), (D), (F) 1 cm. All images taken by M Mizuyama.

development through time as a proxy to clarify the timing of spawning and the possibility of cross-hybridization among putative species.

## MATERIALS AND METHODS

### Specimen collection

Specimens of *Palythoa* species were collected in the intertidal zone from several sites in the Ryukyu Archipelago, including Okinawa-jima Island, Yoron-to Island, Okinoerabu-jima

**Table 1** Characters employed for identification of *Palythoa* species.

| Species | *P. tuberculosa* | *P.* sp. yoron | *P. mutuki* | *P.* aff. *mutuki* |
|---|---|---|---|---|
| Typical environment | Backreef moat - out reef | Reef flat, tide pool | Reef flat, reef edge, surge channel | Reef flat, reef edge, |
| Coenenchyme development | Well-developed | Moderately developed | Not well developed; or stoloniferous | Not well developed; or stoloniferous |
| Polyp structure | immersae (= "embedded") | intermediae (= "moderate") | liberae (= "free-standing") | liberae (="free-standing") |
| Surface structure of capitular ridges | Smooth | Smooth | Jagged | Smooth |
| Number of polyps/colony | >10 | <10 | >10 | >10 |

Island, and Tokunoshima Island (Fig. 1, Table 2) between March 2010 to October 2012. All specimens were stored in 99.5% ethanol for DNA analyses or 5% formalin-SW solution for morphological and anatomical analyses.

Each specimen was identified according to morphological classification methodology (*Pax, 1910*), supplemented with a key to field identification (*Reimer, 2010*), and ecological and morphological aspects of *P.* sp. yoron (*Shiroma & Reimer, 2010*). Characters employed for identification of *Palythoa* species were environment (habitat), coenenchyme development, polyp structure, number of polyps per colony, and numbers of tentacles per polyp. All specimens were identified preliminarily as *Palythoa tuberculosa* (Fig. 2A), *P. mutuki* (Fig. 2B) and *P.* sp. yoron (Fig. 2C). During collection, it was noticed that certain specimens had a similar external appearance with *P. mutuki* but with less well developed marginal ridges and larger polyp sizes. Such specimens were found sympatrically with other specimens, and these were designated as *P.* aff. *mutuki* (Fig. 2D). In addition, spawning timing investigations for all species groups were carried out between June 2010 to December 2010, and from June 2011 to February 2012 at Kaminomine, Tokunoshima, Kagoshima (27°46′09′N, 129°02′16′E) by monthly sampling. In particular, for collecting *P. tuberculosa,* investigation was conducted in a wide area from lagoon tide pools to the outer reef in 2010. However, in 2011–2012 investigations were conducted only in tide pools due to rough sea conditions. At least five different colonies of approximately ten polyps for each species were collected in whole or partially.

## Morphological analyses
### External anatomy

Fixed specimens were cut horizontally at the oral disk height by surgical knife and tweezers under stereomicroscope (S8APO, Leica, Tokyo) and the number of tentacles, which is one of the characters for *Palythoa* species (e.g., *Ryland & Lancaster, 2003*), were counted (Table 3). To eliminate pseudo-replication in comparison among species, a single polyp was chosen with the table of random number from each colony. The mean numbers of tentacles per polyp for each species pair were compared using Mann–Whitney $U$ test with Bonferroni correction.

**Table 2** Examined *Palythoa* specimens in this study from the Ryukyu Archipelago.

| Specimen code | Location/region | GPS code | Species ID | Date (m/d/y) | Collected by | Fixed by | mt COI | mt 16S-rDNA | ITS-rDNA | ALG11 |
|---|---|---|---|---|---|---|---|---|---|---|
| 2PtOkOd | Odo/Okinawa | 1 | *P. tuberculosa* | Aug 18. 09 | MM*1 | 99.5% EtOH | NA | NA | NA | KX389373 |
| 4PtOkOd | Odo/Okinawa | 1 | *P. tuberculosa* | Aug 23. 09 | MM | 99.5% EtOH | NA | KX389335 | NA | KX389374 |
| 5PtOkOd | Odo/Okinawa | 1 | *P. tuberculosa* | Aug 23. 09 | MM | 99.5% EtOH | NA | NA | NA | KX389375 |
| 37PtYoMa | Maehama/Yoron | 2 | *P. tuberculosa* | Mar 03. 10 | JDR*2 | 99.5% EtOH | NA | KX389336 | NA | KX389376 |
| 39PtYoUk | Ukachi/Yoron | 3 | *P. tuberculosa* | Mar 04. 10 | MM | 99.5% EtOH | NA | KX389337 | KX389459 | KX389377 |
| 40PtYoUk | Ukachi/Yoron | 3 | *P. tuberculosa* | Mar 04. 10 | MM | 99.5% EtOH | NA | NA | NA | KX389378 |
| 49PtYoUk | Ukachi(West)/Yoron | 4 | *P. tuberculosa* | Mar 04. 10 | MM | 99.5% EtOH | NA | NA | NA | KX389379 |
| 63PtErYa | Yakomo/Okinoerabu | 5 | *P. tuberculosa* | Mar 05. 10 | MM | 99.5% EtOH | NA | KX389338 | NA | KX389380 |
| 65PtErYa | Yakomo/Okinoerabu | 5 | *P. tuberculosa* | Mar 05. 10 | MM | 99.5% EtOH | NA | NA | NA | KX389381 |
| 91PtToYo | Yonama/Tokunoshima | 6 | *P. tuberculosa* | Mar 08. 10 | MM | 99.5% EtOH | NA | NA | NA | KX389382 |
| 98PtToKa | Kaminomine/Tokunoshima | 7 | *P. tuberculosa* | Mar 09. 10 | MM | 99.5% EtOH | NA | NA | NA | KX389383 |
| 100PtToKa | Kaminomine/Tokunoshima | 7 | *P. tuberculosa* | Mar 09. 10 | MM | 99.5% EtOH | NA | KX389339 | NA | KX389384 |
| 358PtOkAk | Akazaki/Okinawa | 8 | *P. tuberculosa* | Jun 24. 12 | MM | 99.5% EtOH | NA | KX389340 | NA | KX389385 |
| 361PtOkOk | Oku/Okinawa | 9 | *P. tuberculosa* | Jun 25. 12 | MM | 99.5% EtOH | NA | NA | NA | KX389386 |
| 371PtZaAm | Ama/Zamami | 10 | *P. tuberculosa* | Jul 16. 12 | YM*3 | 99.5% EtOH | NA | KX389341 | NA | KX389387 |
| 3PyOkOd | Odo/Okinawa | 1 | *P.* sp. yoron | Aug 18. 09 | MM | 99.5% EtOH | KX389439 | KX389342 | KX389460 | KX389388 |
| 14PyOkOd | Odo/Okinawa | 1 | *P.* sp. yoron | Aug 23. 09 | MM | 99.5% EtOH | KX389440 | KX389343 | KX389472 | KX389389 |
| 15PyOkOd | Odo/Okinawa | 1 | *P.* sp. yoron | Sep 05. 09 | MM | 99.5% EtOH | KX389441 | KX389344 | KX389461 | KX389390 |
| 16PyOkOd | Odo/Okinawa | 1 | *P.* sp. yoron | Sep 05. 09 | MM | 99.5% EtOH | NA | KX389345 | KX389462 | KX389391 |
| 43PyYoUk | Ukachi/Yoron | 3 | *P.* sp. yoron | Mar 04. 10 | MM | 99.5% EtOH | KX389442 | KX389346 | KX389470 | KX389392 |
| 44PyYoUk | Ukachi/Yoron | 3 | *P.* sp. yoron | Mar 04. 10 | MM | 99.5% EtOH | NA | KX389347 | KX389471 | KX389393 |
| 51PyYoUk | Ukachi(West)/Yoron | 4 | *P.* sp. yoron | Mar 04. 10 | MM | 99.5% EtOH | KX389443 | KX389348 | KX389466 | KX389394 |
| 53PyYoUk | Ukachi(West)/Yoron | 4 | *P.* sp. yoron | Mar 04. 10 | MM | 99.5% EtOH | NA | KX389349 | NA | KX389395 |
| 81PyErYa | Yakomo/Okinoerabu | 5 | *P.* sp. yoron | Mar 05. 10 | MM | 99.5% EtOH | KX389444 | KX389350 | KX389463 | KX389396 |
| 83PyErYa | Yakomo/Okinoerabu | 5 | *P.* sp. yoron | Mar 05. 10 | MM | 99.5% EtOH | NA | KX389351 | KX389464 | KX389397 |
| 85PyErYa | Yakomo/Okinoerabu | 5 | *P.* sp. yoron | Mar 05. 10 | MM | 99.5% EtOH | KX389445 | KX389352 | KX389465 | KX389398 |
| 87PyErYa | Yakomo/Okinoerabu | 5 | *P.* sp. yoron | Mar 05. 10 | MM | 99.5% EtOH | NA | KX389353 | NA | KX389399 |
| 105PyToKa | Kaminomine/Tokunoshima | 7 | *P.* sp. yoron | Mar 09. 10 | MM | 99.5% EtOH | KX389446 | KX389354 | KX389467 | KX389400 |
| 107PyToKa | Kaminomine/Tokunoshima | 7 | *P.* sp. yoron | Mar 09. 10 | MM | 99.5% EtOH | KX389447 | KX389355 | KX389468 | KX389401 |
| 109PyToKa | Kaminomine/Tokunoshima | 7 | *P.* sp. yoron | Mar 09. 10 | MM | 99.5% EtOH | NA | KX389356 | KX389469 | NA |
| 359PyOkAk | Akazaki/Okinawa | 8 | *P.* sp. yoron | Jun 24. 12 | MM | 99.5% EtOH | KX389448 | KX389357 | NA | ALG389402 |
| 42PmYoUk | Ukachi/Yoron | 3 | *P. mutuki* | Mar 04. 10 | MM | 99.5% EtOH | NA | KX389366 | KX389488 | KX389403 |
| 61PmYoUk | Ukachi/Yoron | 3 | *P. mutuki* | Mar 04. 10 | JDR | 99.5% EtOH | NA | NA | NA | KX389404 |
| 73PmErYa | Yakomo/Okinoerabu | 5 | *P. mutuki* | Mar 05. 10 | MM | 99.5% EtOH | NA | NA | KX389484 | KX389405 |
| 75PmErYa | Yakomo/Okinoerabu | 5 | *P. mutuki* | Mar 05. 10 | MM | 99.5% EtOH | NA | KX389367 | KX389482 | KX389406 |
| 77PmErYa | Yakomo/Okinoerabu | 5 | *P. mutuki* | Mar 05. 10 | MM | 99.5% EtOH | NA | NA | KX389481 | KX389407 |
| 93PmToYo | Yonama/Tokunoshima | 6 | *P. mutuki* | Mar 08. 10 | MM | 99.5% EtOH | NA | NA | NA | KX389408 |
| 94PmToYo | Yonama/Tokunoshima | 6 | *P. mutuki* | Mar 08. 10 | MM | 99.5% EtOH | NA | NA | NA | KX389409 |
| 95PmToYo | Yonama/Tokunoshima | 6 | *P. mutuki* | Mar 08. 10 | MM | 99.5% EtOH | NA | KX389368 | KX389487 | NA |
| 216PmOkOd | Odo/Okinawa | 1 | *P. mutuki* | May 04. 11 | MM | 99.5% EtOH | NA | KX389369 | NA | KX389410 |
| 218PmOkOd | Odo/Okinawa | 1 | *P. mutuki* | May 04. 11 | MM | 99.5% EtOH | NA | NA | KX389483 | KX389411 |

**Table 2** (*continued*)

| Specimen code | Location/region | GPS code | Species ID | Date (m/d/y) | Collected by | Fixed by | mt COI | mt 16S-rDNA | ITS-rDNA | ALG11 |
|---|---|---|---|---|---|---|---|---|---|---|
| 220PmOkOd | Odo/Okinawa | 1 | *P. mutuki* | May 04. 11 | MM | 99.5% EtOH | NA | NA | KX389489 | KX389412 |
| 222PmOkOd | Odo/Okinawa | 1 | *P. mutuki* | May 04. 11 | MM | 99.5% EtOH | NA | NA | KX389485 | KX389413 |
| 240PmErSu | Sumiyoshi/Okinoerabu | 11 | *P. mutuki* | Jun 18. 11 | MM | 99.5% EtOH | NA | NA | NA | KX389414 |
| 280PmToKa | Kaminomine/Tokunoshima | 7 | *P. mutuki* | Oct 05. 11 | MM | 99.5% EtOH | NA | NA | NA | KX389415 |
| 316PmOkKo | Komesu/Okinawa | 12 | *P. mutuki* ? | Feb 25. 12 | MM | 99.5% EtOH | NA | KX389370 | KX389480 | KX389416 |
| 319PmOkMi | Mizugama/Okinawa | 13 | *P. mutuki* ? | Mar 29. 12 | MM | 99.5% EtOH | NA | KX389371 | KX389486 | KX389417 |
| 320PmOkMi | Mizugama/Okinawa | 13 | *P. mutuki* | Mar 29. 12 | MM | 99.5% EtOH | NA | NA | NA | KX389418 |
| 323PmOkTe | Teniya/Okinawa | 14 | *P. mutuki* | Apr 05. 12 | MM | 99.5% EtOH | NA | NA | NA | KX389419 |
| 324PmOkTe | Teniya/Okinawa | 14 | *P. mutuki* | Apr 05. 12 | MM | 99.5% EtOH | NA | NA | NA | KX389420 |
| 349PmOkSh | Shioya Bay/Okinawa | 15 | *P. mutuki* | Jun 17. 12 | MM | 99.5% EtOH | NA | NA | NA | KX389421 |
| 362PmOkOk | Oku/Okinawa | 9 | *P. mutuki* | Jun 25. 12 | MM | 99.5% EtOH | NA | NA | NA | KX389422 |
| 155PamErYa | Yakomo/Okinoerabu | 5 | *P.* aff. *mutuki* | July 25. 10 | MM | 70% EtOH | KX389449 | KX389358 | KX389473 | KX389423 |
| 159PamToKa | Kaminomine/Tokunoshima | 7 | *P.* aff. *mutuki* | July 28. 10 | MM | 70% EtOH | NA | NA | NA | KX389424 |
| 229PamErYa | Yakomo/Okinoerabu | 5 | *P.* aff. *mutuki* | Jun 17. 11 | MM | 99.5% EtOH | KX389450 | KX389359 | KX389474 | NA |
| 231PamErYa | Yakomo/Okinoerabu | 5 | *P.* aff. *mutuki* | Jun 17. 11 | MM | 99.5% EtOH | KX389451 | KX389360 | KX389475 | KX389425 |
| 233PamErYa | Yakomo/Okinoerabu | 5 | *P.* aff. *mutuki* | Jun 17. 11 | MM | 99.5% EtOH | KX389452 | KX389361 | KX389476 | KX389426 |
| 237PamErSu | Sumiyoshi/Okinoerabu | 11 | *P.* aff. *mutuki* | Jun 18. 11 | MM | 99.5% EtOH | KX389453 | KX389362 | KX389479 | NA |
| 248PamToKa | Kaminomine/Tokunoshima | 7 | *P.* aff. *mutuki* | Jun 21. 11 | MM | 99.5% EtOH | KX389454 | KX389363 | KX389478 | KX389427 |
| 250PamToKa | Kaminomine/Tokunoshima | 7 | *P.* aff. *mutuki* | Jun 21. 11 | MM | 99.5% EtOH | KX389455 | KX389364 | KX389477 | KX389428 |
| 328PamOkTe | Teniya/Okinawa | 14 | *P.* aff. *mutuki* | Apr 05. 12 | MM | 99.5% EtOH | KX389456 | NA | NA | KX389429 |
| 364PamOkOk | Oku/Okinawa | 9 | *P.* aff. *mutuki* | Jun 25. 12 | MM | 99.5% EtOH | KX389457 | KX389365 | NA | KX389430 |
| 215PsOkIk | Ikei E/Okinawa | 16 | *Palythoa* sp. sakurajimensis | Apr 29. 11 | MM | 99.5% EtOH | NA | KX389372 | KX389491 | KX389431 |
| 1595[a] | Wanli Tung/Taiwan | 2 | *Palythoa* sp. sakurajimensis | Sep. 09 | JDR | 99.5% EtOH | KF499697 | KF499661 | KX389490 | KX389432 |
| 1597[a] | Wanli Tung/Taiwan | 1 | *Palythoa* sp. sakurajimensis | Sep. 09 | JDR | 99.5% EtOH | KF499696 | KF499662 | KF499778 | KX389433 |
| 1635[a] | Bitouchiao/Taiwan | 8 | *Palythoa* sp. sakurajimensis | Sep. 09 | JDR | 99.5% EtOH | KF499735 | KF499652 | KF499783 | KX389434 |
| 321PhOkMi | Mizugama/Okinawa | 13 | *P. heliodiscus* | Mar 29. 12 | MM | 99.5% EtOH | KX389458 | NA | NA | KX389435 |
| TN116 | Mizugama/Okinawa | 13 | *P. heliodiscus* | Aug 19. 10 | TN*4 | 99.5% EtOH | NA | NA | NA | KX389436 |
| TN119 | Mizugama/Okinawa | 13 | *P. heliodiscus* | Jul 4. 12 | TN | 99.5% EtOH | NA | NA | NA | KX389437 |
| TN121 | Mizugama/Okinawa | 13 | *P. heliodiscus* | Jul 4. 12 | TN | 99.5% EtOH | NA | NA | NA | KX389438 |

**Notes.**

MM*[1], Masaru Mizuyama; JDR*[2], James Davis Reimer; YM*[3], Yu Miyazaki; TN*[4], Tohru Nishimura.

[a]Specimen from *Reimer et al. (2013)*.

GPS code: **1**, N26°05′15″, E127°42′30″; **2**, N27°01′16″, E128°26′28″; **3**, N27°04′00″, E128°25′24″, **4**, N27°03′54″, E128°25′11″; **5**, N27°20′05″, E128°32′49″; **6**, N27°52′17″, E128°53′23″; **7**, N27°46′13″, E129°02′18″; **8**, N26°49′17″, E128°18′50″; **9**, N26°50′49″, E128°17′12″; **10**, N26°13′35″, E127°17′33″; **11**, N27°21′21″, E128°31′44″; **12**, N26°05′17″, E127°42′06″; **13**, N26°21′35″, E127°44′20″; **14**, N26°34′07″, E128°08′48″; **15**, N26°39′50″, E128°06′31″; **16**, N26°23′40″, E128°00′22″.

### Cnidae

Cnidae analyses were conducted using undischarged nematocysts from the tentacles, column, pharynx, and mesenteriel filaments of polyps ($n = 3$/species group) under a Nikon Eclipse80i stereomicroscope (Nikon, Tokyo). Cnidae sizes were measured using ImageJ v1.45s (*Rasband, 2012*). Cnidae classification followed *England (1991)* and *Ryland & Lancaster* (*2004*; see also Table 4).

### Spawning period investigation

Ovary development of all preserved colonies was observed via cross sections made by cutting polyps vertically through the mouth located at the center of oral disk under a

**Table 3** The mean number of tentacles ± standard deviation and results of Mann–Whitney *U* test with Bonferroni correction between each *Palythoa* species pairs. *N* = total number of examined polyps for each species (one per colony).

| Species | *P. tuberculosa* | *P.* sp. yoron | *P. mutuki* | *P.* aff. *mutuki* |
|---|---|---|---|---|
| | | | Mann–Whitney *U* test | |
| *P. tuberculosa* | $31.6 \pm 3.4$ ($N = 11$) | <0.001 | <0.001 | <0.001 |
| *P.* sp. yoron | | $40.5 \pm 2.56$ ($N = 8$) | <0.001 | <0.001 |
| *P. mutuki* | | | $54.4 \pm 7.43$ ($N = 7$) | <0.001 |
| *P.* aff. *mutuki* | | | | $71 \pm 4.14$ ($N = 8$) |

stereomicroscope. During anthozoans' oogenesis, oocytes form a single-layered germinal ribbon down the mesoglea of the central third of the septa. Subsequently, the germinal ribbon develops a sequence of swollen nodes where the septum folds locally in an S and the layers fuse (*Ryland, 1997*; *Ryland, 2000*). When we observed a germinal ribbon in a polyp, we counted the polyp as "possessing developing ovaries", and the number of polyps possessing developing ovaries were totaled. To evaluate the spawning period of each species, the ratio of the number of polyps possessing developing and/or developed ovaries to the total number of polyps examined was calculated over time. When the calculated proportion of developed/developing ovaries dropped dramatically, we designated this as the start of the estimated spawning period. The end of the estimated spawning period was defined as the point where the number of developed/developing ovaries reached 0%.

## Molecular analyses
### DNA extraction, PCR amplification and direct sequencing

DNA from each specimen was extracted using a DNeasy Blood and Tissue Kit (QIAGEN, Tokyo, Japan) according to the manufacturer's instructions. A small amount of tissue from each specimen was removed using a surgical knife sterilized by open flame. Extracted DNA was subsequently stored at −20 °C, and then we amplified target sequences via polymerase chain reaction (PCR).

Three molecular markers that have previously been used for differentiation of *Palythoa* were chosen; (1) the mitochondrial 16S of ribosomal DNA (mt 16S-rDNA), (2) the mitochondrial cytochrome c oxidase subunit I (mtCOI), and (3) the internal transcribed spacer region of nuclear ribosomal DNA (ITS-rDNA) (*Reimer et al., 2004*; *Sinniger et al., 2005*; *Reimer et al., 2007a*, etc.). Furthermore, a nuclear housekeeping gene, (4) asparagine-linked glycosylation 11 protein (ALG11) region, was also examined for the first time in zoantharians. This marker has been found to be more informative than mtCOI in examining sponge relationships and succeeded in solving previously debated nodes (*Hill et al., 2013*) and has also been considered to be useful for resolving cnidarian relationships (*Belinky et al., 2012*).

Mizuyama et al. (2018), *PeerJ*, DOI 10.7717/peerj.5132

Peer J

**Table 4  Cnidae types and sizes of *Palythoa* aff. *mutuki*, *Palythoa mutuki*, *Palythoa* sp. yoron and *Palythoa tuberculosa*.** Frequency: relative abundance of cnidae type in decreasing order; numerous, common, occasional, rare, very rare ($N$ = number of specimens found/total specimens examined).

| | *Palythoa* aff. *mutuki* | | *Palythoa mutuki* | | *Palythoa* sp. yoron | | *Palythoa tuberculosa* | |
|---|---|---|---|---|---|---|---|---|
| | Length × width (μm) | Frequency | Length × width (μm) | Frequency | Length × width (μm) | Frequency | Length × width (μm) | Frequency |
| **Tentacles** | | | | | | | | |
| Spirocysts | 12–36 × 3–8 | Numerous (3/3) | 13–41 × 2–8 | Numerous (3/3) | 11–36 × 2–6 | Common (3/3) | 17–37 × 3–7 | Numerous (3/3) |
| Basitrichs | 16–55 × 4–7 | Common (3/3) | 14–63 × 3–8 | Numerous (3/3) | 25–73 × 2–9 | Numerous (3/3) | 25–37 × 4–6 | Common (3/3) |
| Holotrichs small | 15–20 × 5–9 | Rare (1/3) | – | 0 | – | 0 | – | 0 |
| Holotrichs large | 35–77 × 19–31 | Occasional (2/3) | 39–78 × 18–32 | Numerous (3/3) | 47–82 × 21–34 | Numerous (3/3) | 28–85 × 17–37 | Occasional (2/3) |
| P-mastigophores | 25–50 × 5–10 | Common (3/3) | 15 × 4 | Very rare (single specimen) | 26–29 × 5–6 | Occasional (2/3) | 46–51 × 6–8 | Rare (1/3) |
| **Column** | | | | | | | | |
| Spirocysts | – | 0 | – | 0 | – | 0 | 16–34 × 3–6 | Rare (1/3) |
| Basitrichs | 21–53 × 5–7 | Occasional (2/3) | 25–83 × 5–9 | Common (3/3) | – | 0 | 25–69 × 4–10 | Common (3/3) |
| Holotrichs small | 21 × 7 | Very rare (single specimen) | 19–24 × 8 | Rare (1/3) | – | 0 | – | 0 |
| Holotrichs large | 32–69 × 15–30 | Numerous (3/3) | 24–85 × 17–31 | Numerous (3/3) | 39–88 × 18–36 | Numerous (3/3) | 34–81 × 14–38 | Numerous (3/3) |
| P-mastigophores | 21–46 × 6–8 | Rare (1/3) | – | 0 | – | 0 | 52–54 × 7–8 | Occasional (2/3) |
| **Actinopharynx** | | | | | | | | |
| Spirocysts | – | 0 | 18–32 × 4–6 | Occasional (2/3) | 16–65 × 3–8 | Occasional (2/3) | 19–36 × 4–7 | Rare (1/3) |
| Basitrichs | 19–55 × 4–10 | Numerous (3/3) | 16–72 × 3–8 | Numerous (3/3) | 17–69 × 3–9 | Numerous (3/3) | 22–62 × 3–10 | Numerous (3/3) |
| Holotrichs small | 19–20 × 7–8 | Rare (1/3) | – | 0 | – | 0 | – | 0 |
| Holotrichs large | 34–93 × 18–33 | Numerous (3/3) | 34–72 × 4–31 | Numerous (3/3) | 38–77 × 10–33 | Common (3/3) | 40–85 × 18–38 | Numerous (3/3) |
| P-mastigophores | 29–40 × 7–11 | Rare (1/3) | – | 0 | 21–29 × 6–7 | Occasional (2/3) | 28–52 × 5–8 | Rare (1/3) |
| **Mesenteries filaments** | | | | | | | | |
| Spirocysts | 15 × 24 | Very rare (single specimen) | – | 0 | – | 0 | 28 × 8 | Very rare (single specimen) |
| Basitrichs | 25–69 × 4–10 | Numerous (3/3) | 41–80 × 5–10 | Numerous (3/3) | 33–66 × 4–9 | Numerous (3/3) | 24–74 × 5–9 | Numerous (3/3) |
| Holotrichs small | – | 0 | – | 0 | – | 0 | – | 0 |
| Holotrichs large | 37–64 × 22–35 | Numerous (3/3) | 44–83 × 21–32 | Numerous (3/3) | 51–90 × 21–35 | Numerous (3/3) | 45–85 × 22–42 | Numerous (3/3) |
| P-mastigophores | 27–39 × 5–10 | Occasional 2/3 | 21 × 6 | Very rare (single specimen) | 21–29 × 4–8 | Common (3/3) | 21–57 × 5–11 | Occasional (2/3) |

Thermal cycler programs were set to the following conditions: (1) mt 16S-rDNA; an initial denaturing step at 94 °C for 2 min, followed by 40 cycles of 30 s 94 °C, 1 min annealing at 52 °C and 2 min extension at 72 °C, followed by 5 min final elongation at 72 °C with Zoantharia-specific primer set 16Sant1a (5′-GCC ATG AGT ATA GAC GCA CA-3′) and 16SbmoH (5′-CGA ACA GCC AAC CCT TGG-3′) (*Sinniger et al., 2005*); (2) mtCOI; 1 min at 95 °C, then 35 cycles: 1 min at 95 °C, 1 min at 40 °C and 90 s at 72 °C, followed by 7 min at 72 °C with the universal primers HCO2198 (5′-TAA ACT TCA GGG TGA CCA AAA AAT CA-3′) and LCO1490 (5′-TAA ACT TCA GGG TGA CCA AAA AAT CA-3′) (*Folmer et al., 1994*); and (3) ITS-rDNA; 1 min at 95 °C , then 35 cycles of 1 min at 94 °C, 1 min at 50 °C, and 2 min at 72 °C, followed by 10 min at 72 °C with Zoantharia-specific primers Zoan-f (5′-CTT GAT CAT TTA GAG GGA GT-3′) and Zoan-r (5′-CGG AGA TTT CAA ATT TGA GCT-3′) (*Reimer et al., 2007a*).

Amplification for the remaining coding region (ALG11) was performed by touch-down PCR and nested PCR because of low numbers of copies in the whole genome as this is a single-copy gene. For ALG11, although we basically followed the original protocols (*Sperling, Pisani & Peterson, 2007*; *Belinky et al., 2012*), some modifications were required to fit the thermal cycler we used, and the conditions were as follows: (4) ALG11 first touchdown, 2 min at 95 °C, then 13 cycles of 1 min at 95 °C, 1 min at 52-40 °C (dropping one degree for each cycle), 1.5 min at 72 °C; followed by 20 cycles of 1 min at 95 °C, 1 min at 52 °C, 1.5 min at 72 °C; lastly 5 min at 72 °C with primers ALG11-D1 (5′-TTY CAY CCN TAY TGY AAY GCN GGN GG-3′) and ALG11-R1 (5′-ATN CCR AAR TGY TCR TTC CAC AT-3′), and (5) MAT-f (5′-GGN GAR GGN CAY CCN GAY AA-3′). In the second touchdown procedure an amplicon of the first touchdown was utilized as the template, followed by 2 min at 95 °C, then 35 cycles of 1 min at 95 °C, 1 min at 52 °C, and 1.5 min at 72 °C. In the end, nested PCR was performed with 2 min at 95 °C, and then 35 cycles of 1 min at 95 °C, 1 min at 52 °C, and 1.5 min at 72 °C with primers ALG11-D2 (5′-TGY AAY GCN GGN GGN GGN GGN GA-3′) and ALG11-R2 (5′-CCR AAR TGY TCR TTC CAC ATN GTR TG-3′).

Amplicons were outsourced for sequencing to a private sequencing company (Fasmac Co., Ltd., Kanagawa, Japan) on an Applied Biosystems 3730xl DNA sequencer, using BigDye Terminator V3.1 and the same primer sets as for PCR as described above. Sequence data were edited using BioEdit v.7.2.0 (*Hall, 1999*).

## Sequence alignment

The total number of novel sequences obtained from specimens in this study were (1) mt 16S-rDNA; 38; (2) mtCOI; 20; (3) ITS-rDNA; 35 and (4) ALG11; 65, respectively. Obtained sequences were aligned by BioEdit v7.2.0 (*Hall, 1999*) with other sequences deposited in GenBank (Table 5).

As numerous indels (inserts and deletions) were confirmed in ITS-rDNA sequences, alignment was performed using ClustalW (*Thompson, Higgins & Gibson, 1994*) with gap penalties of 10 for open and 1 for extended, followed by manual fixing for obviously misaligned areas such as gap position. Sequences of the 5.8S rDNA region located between internal transcribed spacer 1 (ITS1) and internal transcribed spacer 2 (ITS2) were removed

**Table 5** GenBank accession numbers of genus *Palythoa* sequences used in this study.

| Sequence code | Species | mtCOI accession number | mt 16S-rDNA accession number | ITS-rDNA accession number | Reference |
|---|---|---|---|---|---|
| PtEW3 | *P. tuberculosa* | NA | NA | DQ997902 | *Reimer et al. (2007a)* |
| PtAT1 | *P. tuberculosa* | AB219195 | NA | NA | *Reimer, Takishita & Maruyama (2006)* |
| PtAT2 | *P. tuberculosa* | AB219196 | NA | DQ997897 | *Reimer, Takishita & Maruyama (2006)* |
| PtBA1 | *P. tuberculosa* | AB219197 | NA | NA | *Reimer, Takishita & Maruyama (2006)* |
| PtWK1 | *P. tuberculosa* | AB219198 | NA | NA | *Reimer, Takishita & Maruyama (2006)* |
| PtYS1 | *P. tuberculosa* | AB219200 | NA | NA | *Reimer, Takishita & Maruyama (2006)* |
| PtMil1 | *P. tuberculosa* | AB219199 | AB219218 | NA | *Reimer, Takishita & Maruyama (2006)* |
| PtIsK3 | *P. tuberculosa* | AB219203 | NA | NA | *Reimer, Takishita & Maruyama (2006)* |
| PtEO1 | *P. tuberculosa* | AB219205 | NA | NA | *Reimer, Takishita & Maruyama (2006)* |
| PtKK1 | *P. tuberculosa* | AB219206 | NA | NA | *Reimer, Takishita & Maruyama (2006)* |
| PtIsK2 | *P. tuberculosa* | AB219207 | NA | NA | *Reimer, Takishita & Maruyama (2006)* |
| PtYS4 | *P. tuberculosa* | NA | NA | DQ997903 | *Reimer, Takishita & Maruyama (2006)* |
| PtIrHo16 | *P. tuberculosa* | NA | NA | DQ997909 | *Reimer, Takishita & Maruyama (2006)* |
| PtCN1 | *P. tuberculosa* | NA | NA | DQ997896 | *Reimer, Takishita & Maruyama (2006)* |
| PtCN14 | *P. tuberculosa* | NA | NA | DQ997933 | *Reimer, Takishita & Maruyama (2006)* |
| PtIsO1 | *P. tuberculosa* | AB219202 | NA | NA | *Reimer, Takishita & Maruyama (2006)* |
| PtIsO13 | *P. tuberculosa* | NA | NA | DQ997919 | *Reimer, Takishita & Maruyama (2006)* |
| PtIsO11 | *P. tuberculosa* | NA | NA | DQ997929 | *Reimer, Takishita & Maruyama (2006)* |
| PtIsrael13 | *P. tuberculosa* | NA | NA | DQ997931 | *Reimer, Takishita & Maruyama (2006)* |
| PtOtsFu11 | *P. tuberculosa* | NA | NA | DQ997945 | *Reimer et al. (2007a)* |
| PtIrHo11 | *P. tuberculosa* | NA | NA | DQ997914 | *Reimer et al. (2007a)* |
| PtOtsNi3 | *P. tuberculosa* | NA | NA | DQ997939 | *Reimer, Takishita & Maruyama (2006)* |
| PtIrHo13 | *P. tuberculosa* | NA | NA | DQ997911 | *Reimer et al. (2007a)* |
| PtL1 | *P. tuberculosa* | NA | EU333661 | NA | *Reimer & Todd (2009)* |
| PtK2 | *P. tuberculosa* | NA | EU333654 | NA | *Reimer & Todd (2009)* |
| PtL3 | *P. tuberculosa* | NA | EU333662 | NA | *Reimer & Todd (2009)* |
| PtK7 | *P. tuberculosa* | NA | EU333657 | NA | *Reimer & Todd (2009)* |
| PtYoS1 | *P. sp. yoron* | AB219204 | AB219219 | DQ997921 | *Reimer et al. (2007a)* |
| PmAT | *P. mutuki* | AB219209 | NA | NA | *Reimer, Takishita & Maruyama (2006)* |
| PmPM2 | *P. mutuki* | AB219210 | NA | NA | *Reimer, Takishita & Maruyama (2006)* |
| Pm1162 | *P. mutuki* | JF419796 | NA | NA | *Reimer et al. (2011)* |
| Pm1163 | *P. mutuki* | JF419788 | NA | NA | *Reimer et al. (2011)* |
| PmBA1 | *P. mutuki* | AB219215 | NA | NA | *Reimer, Takishita & Maruyama (2006)* |
| PmYS1 | *P. mutuki* | AB219213 | NA | NA | *Reimer, Takishita & Maruyama (2006)* |
| PmIrHo1 | *P. mutuki* | NA | NA | DQ997888 | *Reimer et al. (2007a)* |
| PmYS2 | *P. mutuki* | NA | NA | DQ997892 | *Reimer et al. (2007a)* |
| PpAT1 | *P. mutuki* | AB219211 | AB219220 | DQ997891 | *Reimer et al. (2007a)* |
| PmMil1 | *P. mutuki* | AB219217 | AB219225 | DQ997889 | *Reimer et al. (2007a)* |
| PmEs1 | *P. mutuki* | NA | NA | DQ997894 | *Reimer et al. (2007a)* |
| PpAT2 | *P. mutuki* | AB219212 | AB219221 | NA | *Reimer, Takishita & Maruyama (2006)* |

**Table 5** (*continued*)

| Sequence code | Species | mtCOI accession number | mt 16S-rDNA accession number | ITS-rDNA accession number | Reference |
|---|---|---|---|---|---|
| PpYS1 | *P. mutuki* | NA | AB219222 | NA | *Reimer, Takishita & Maruyama (2006)* |
| PamTOB51 | *P.* aff. *mutuki* | NA | GQ464873 | GQ464902 | *Swain (2010)* |
| PsPSH1 | *P.* sp. sakurajimensis | NA | DQ997842 | DQ997886 | *Reimer et al. (2007a)* |
| PsPWS1 | *P.* sp. sakurajimensis | NA | DQ997863 | DQ997887 | *Reimer et al. (2007a)* |
| PsPEWn1 | *P.* sp. sakurajimensis | NA | DQ997862 | NA | *Reimer et al. (2007a)* |
| PsGYi | *P.* sp. sakurajimensis | KF499720 | NA | NA | *Reimer et al. (2013)* |
| Ps1595 | *P.* sp. sakurajimensis | KF499697 | NA | KX389490 | *Reimer et al. (2013)* |
| Ps1597 | *P.* sp. sakurajimensis | KF499696 | NA | KF499778 | *Reimer et al. (2013)* |
| Ps1635 | *P.* sp. sakurajimensis | KF499735 | NA | KF499776 | *Reimer et al. (2013)* |
| PhIsK2 | *P. heliodiscus* | NA | NA | DQ997885 | *Reimer et al. (2007a)* |
| PhIsK11 | *P. heliodiscus* | NA | NA | DQ997880 | *Reimer et al. (2007a)* |
| PhEK1 | *P. heliodiscus* | NA | NA | DQ997882 | *Reimer et al. (2007a)* |
| PhSaiLL1 | *P. heliodiscus* | AB219214 | AB219223 | NA | *Reimer, Takishita & Maruyama (2006)* |
| PhEK1 | *P. heliodiscus* | NA | AB219224 | NA | *Reimer, Takishita & Maruyama (2006)* |
| PhPpM1 | *P. heliodiscus* | AB219216 | NA | NA | *Reimer, Takishita & Maruyama (2006)* |

from analyses because the substitution rate is apparently lower than ITS1 and ITS2, and an admixture of different substitution rates could lead to a misleading choice of the appropriate substitution model. Additionally, in order to not overestimate for genetic distance in following phylogenetic analyses, sites were removed if they had a percentage of gaps and/or ambiguous sites higher than 95% (partial-deletion option).

Fifty-six out of sixty-five specimens had one or more degenerate codes in sequences of the ALG11 region. All degenerate codes were divided into two standard bases using PHASE v2.1.1, which implements a Bayesian statistical method for reconstructing haplotypes from population genotype data (*Stephens, Smith & Donnelly, 2001*; *Stephens & Scheet, 2005*). Furthermore, first and second codon positions were removed from the dataset by checking amino acid sequences after translation.

Thus, each dataset was modified as needed, with additional previously reported sequences added from GenBank, and we generated four alignments; (1) mtCOI; 451 bp of 47 sequences; (2) mt 16S-rDNA; 697 bp of 54 sequences; (3) ITS-rDNA; 317 bp of 60 sequences and (4) ALG11; 578 bp of 121 sequences. These were used for subsequent phylogenetic analyses.

## Substitution model selection

Substitution models for each gene were estimated by jModelTest v2.1.3 (*Darriba et al., 2012*) through the following steps. Initially, likelihood calculations were carried out for all substitution models with configurations of seven substitution schemes, equal or unequal base frequencies (+F), rate variation among sites with a number of rate categories (+G, nCat 5) and base tree topology (ML optimized). Subsequently, the most appropriate model for each marker was selected under (i) the corrected Akaike information criterion (AICc) for Maximum-Likelihood and neighbor-joining phylogenetic estimation, or (ii) Bayesian

information criterion (BIC) for Bayes estimation. Thus, the (i)TrN/(ii)TrNef for mt 16S-rDNA, (i)F81/(ii)JC for mtCOI, (i,ii)K80+Γ for ITS-rDNA, and (i)K80+Γ/(ii)TPM1uf+Γ models for ALG11 were employed, respectively.

## Gene tree estimations

For four distinct datasets (mt 16S-rDNA, mtCOI, ITS-rDNA, ALG11), phylogenetic analyses were applied independently with the optimal substitution model under AICc estimated by jModelTest. Maximum-Likelihood (ML) analyses were performed using PhyML (*Guindon & Gascuel, 2003*) and neighbor-joining (NJ) methods were performed using MEGA5.2.2 (*Tamura et al., 2011*). All other parameters besides substitution model and the discrete gamma distribution were implemented with the default value. Bootstrap analyses (*Felsenstein, 1985*) of 1,000 replicates were tested to evaluate the support of every branch.

Bayesian inference for gene trees was performed using BEAST v.1.7.0 (*Drummond et al., 2010*) with the optimal substitution model under BIC. All parameters were used as default values except for the molecular clock, in which the rate was changed to the log-normal relaxed model, while only the substitution model for ALG11 was modified to TPM1uf after generating the initial setting file. Four Markov chain Monte Carlo (MCMC) simulations were run for 10 million generations with sampling intervals of 1,000. Convergence of analyses and adequacy of the sample sizes, with ESS values above 200 (ESS = the number of effectively independent draws from the posterior distribution that the Markov chain is equivalent to) were confirmed in Tracer v.1.5. (*Rambaut et al., 2013*). Analyses were combined using LogCombiner v.1.8.0, which is included within BEAST, after excluding the first 10% as burn-in. Obtained trees were summarized in a maximum clade credibility tree using TreeAnotator v.1.8.0 and visualized in FigTree v.1.4.0.

## Species tree estimations

*BEAST estimates the species tree directly from the sequence data, nucleotide substitution model parameters and the coalescent process (*Heled & Drummond, 2010*). The species trees were built by grouping all 235 sequences by putative species groups and simultaneously estimating each of three individual gene trees (mt 16S-rDNA, ITS-rDNA and ALG11), and the summary species trees using BEAST were drawn for two different species model; (1) a six species model including *P. tuberculosa*, *P.* sp. yoron, *P. mutuki*, *P.* aff. *mutuki*, *P.* sp. sakurajimensis sensu *Reimer et al. (2007a)* and *Reimer et al. (2007b)* and *P. heliodiscus*, and (2) a four species model combining *P.* sp. yoron with *P. tuberculosa*, and *P.* aff. *mutuki* with *P. mutuki*, along with *P.* sp. sakurajimensis and *P. heliodiscus*.

All parameters were used as default except for; (1) the molecular clock rate, which was changed to the log-normal relaxed model (*Drummond et al., 2006*), (2) the substitution rate for mt 16S-rDNA, for which the range was calibrated to between 0.001-0.002/Mya based on the reported substitution rate for mtCOI (*Shearer et al., 2002*), and (3) the substitution model for ALG11 was modified to TPM1uf after generating the setting file. MCMC analyses were run for 100 million generations with sampling intervals of 10,000 and excluding the first 10% as burn-in. All the parameters in the output file were confirmed in Tracer v1.5. Obtained trees were summarized in a maximum clade credibility tree using TreeAnotator v.1.8.0.
## RESULTS

### Morphological analyses

The numbers of tentacles were measured for single randomly selected polyps from eleven colonies of *P. tuberculosa*, eight colonies of *P.* sp. yoron, seven colonies of *P. mutuki,* and eight colonies of *P.* aff. *mutuki*. The mean number of tentacles ± standard deviation per polyp was 31.6 ± 3.4 for *P. tuberculosa*, 40.5 ± 2.6 for *P.* sp. yoron, 54.4 ± 7.4 for *P. mutuki,* and 71.0 ± 4.1 for *P.* aff. *mutuki*. Each respective mean number of tentacles was significantly different ($p < 0.01$) from all others in all pair tests (Table 3).

For cnidae, many subtle differences in sizes of the various types of cnidae present in different tissues were present (Table 4; Fig. 3). However, the most obvious differences were in small holotrichs, which were rarely observed in the tentacles of column of both *P.* aff. *mutuki* and *P. mutuki*, and additionally observed in the tentacles and pharynx of *P.* aff. *mutuki*, but were never observed in tissues of *P.* sp. yoron or *P. tuberculosa* (Table 4). However, these small holotrichs were only observed in one out of three specimens each of *P.* aff. *mutuki* and *P. mutuki*, and thus no diagnostic differences were observed in the cnidae of all four species-groups examined (Table 4).

In summary, we could clearly distinguish all four *Palythoa* species groups based on tentacle numbers (Table 3), as well as gross external morphology (Fig. 2), but not via cnidae analyses (Table 4).

### Estimated spawning period

During the initial investigation of June to December in 2010, developed ovaries were observed in *P. tuberculosa* from the middle of June to the middle of September with decreasing numbers of polyps possessing ova (Fig. 4A, Table 6). Additionally, matured eggs were also observed multiple times (on 28 July and 20 September). In contrast, developed ovaries and matured eggs were observed (Figs. 5A, 5B) only one time (on 26 October) in *P.* sp. yoron. As well, developing ovaries were observed in *P. mutuki* from the end of July to the middle of September, however, no matured eggs were observed during this investigation.

In 2011, developed ovaries were observed in *P.* aff. *mutuki* on 15 June (Figs. 5E, 5F, 4B), and subsequently developed ovaries were observed in *P.* sp. yoron in early October and early November (Figs. 5C, 5D), for the second consecutive year. On the other hand, no fully developed ovaries were observed in *P. tuberculosa* and *P. mutuki* despite developing ovaries being observed continuously during the summer season (on 23 July, 22 August and 5 October), similar as observed in 2010.

### Phylogenetic analyses
#### Molecular phylogenetic trees

*mtCOI.* The phylogenetic tree resulting from maximum likelihood analyses of the mtCOI sequence alignment is shown in Fig. 6A. *Palythoa tuberculosa*, *P.* sp. yoron, *P. mutuki* and *P.* aff. *mutuki* formed one mixed clade with low bootstrap support (Maximum-Likelihood [ML] ≤ 50%, Neighbor-joining [NJ] = 64%, Bayes [B] = 0.99). Three sequences of

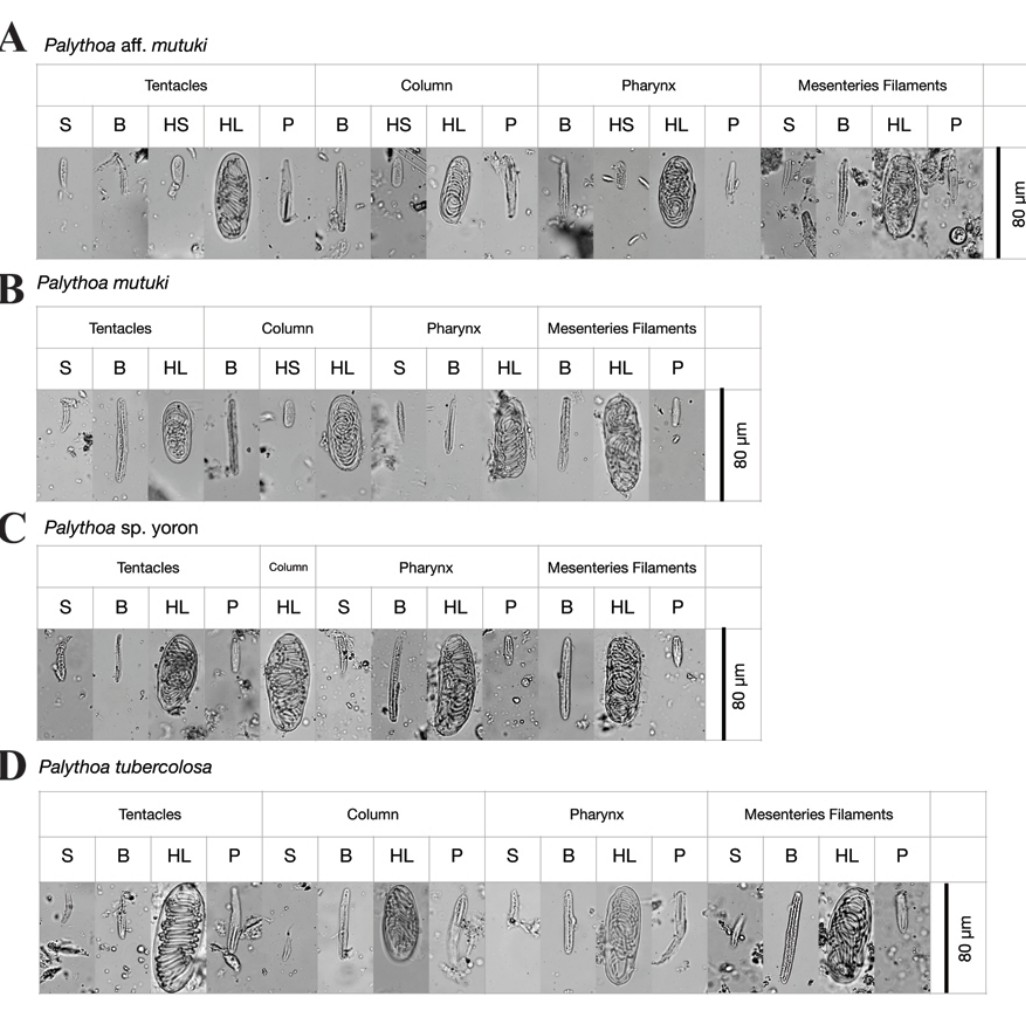

**Figure 3  Cnidae of *Palythoa* species examined in this study.** Cnidae in tentacles, column, pharynx, and filaments of (A) *Palythoa* aff. *mutuki*, (B) *Palythoa mutuki*, (C) *Palythoa* sp. yoron, and (D) *Palythoa tuberculosa*. S, spirocysts; B, basitrichs; HS, holotrichs small; HL, holotrichs large; P, microbasic p-mastigophores.

*P. mutuki* used in previous research (*Reimer et al., 2007a*; *Reimer et al., 2007b*; *Reimer et al., 2011*) formed one group with sequences from *P.* sp. sakurajimensis.

*mt 16S-rDNA.* The phylogenetic tree resulting from maximum likelihood analyses of the mt 16S-rDNA sequence alignment is shown in Fig. 6B. *Palythoa tuberculosa, P.* sp. yoron, *P. mutuki* and *P.* aff. *mutuki* formed one mixed clade with low bootstrap support (ML = 65%, NJ = 64%, B < 0.50). Within this mixed clade, *P. mutuki* and *P.* aff. *mutuki* formed a mixed subclade with low bootstrap support in ML and NJ analyses, however, this monophyletic clade was strongly supported in Bayesian analyses (ML = 64%, NJ = 64%, B = 1.0). Additionally, two sequences of *P. mutuki* from GenBank that were distinguished from other sequences of *P. mutuki* in previous research (*Reimer, Takishita & Maruyama,*

A

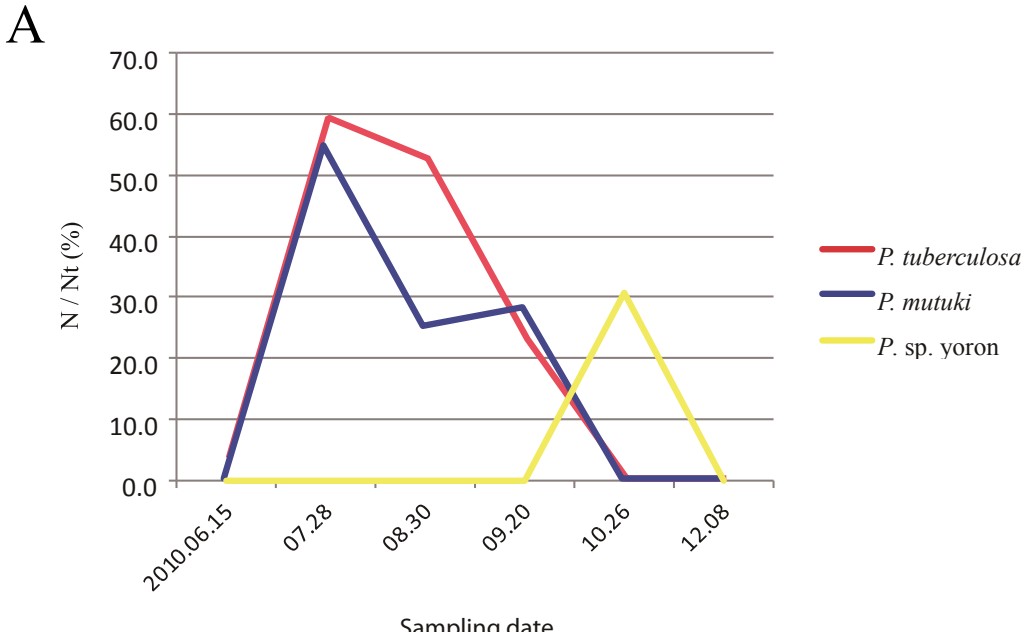

B

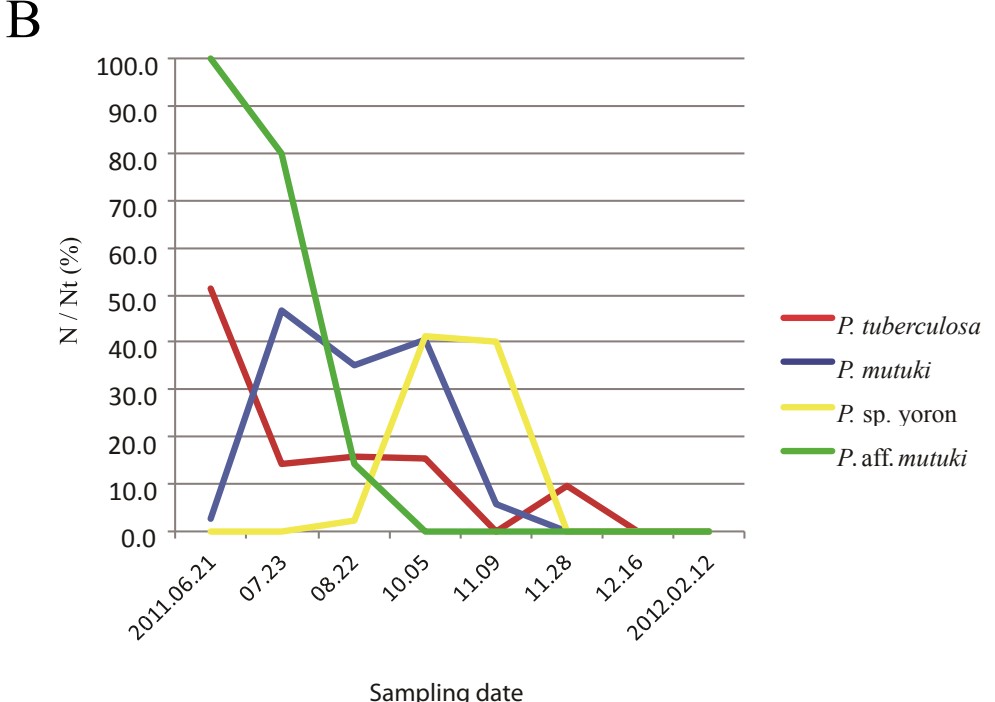

**Figure 4 Monthly change of ratio of number of polyps possessing developing and/or developed ovaries (N) on total number of examined polyps (Nt).** (A) Monthly change of ratio of number of polyps possessing developing and/or developed ovaries (N) on total number of examined polyps (Nt) in 2010. Red, *P. tuberculosa*; blue, *P. mutuki*; yellow, *P.* sp. yoron. (B) Monthly change of ratio of number of polyps possessing developing and/or developed ovaries (N) on total number of examined polyps (%) in 2011. Red, *P. tuberculosa*; blue, *P. mutuki*; yellow, *P.* sp. yoron; green, *P.* aff. *mutuki*.

**Table 6 Ovary development in polyps of four species of *Palythoa*.** Number of polyps possessing developing and/or developed ovaries (N), total number of examined polyps (Nt) and ratio of N to Nt for collected specimens of *P. tuberculosa*, *P. mutuki*, *P.* sp. yoron and *P.* aff. *mutuki* on each sampling date.

| Species | *P. tuberculosa* | | | *P. mutuki* | | | *P.* sp. yoron | | | *P.* aff. *mutuki* | | |
|---|---|---|---|---|---|---|---|---|---|---|---|---|
| Date | N | Nt | N/Nt (%) | N | Nt | N/Nt (%) | N | Nt | N/Nt (%) | N | Nt | N/Nt (%) |
| 2010.06.15 | 2 | 60 | 3 | 0 | 13 | 0 | 0 | 18 | 0 | – | – | – |
| 07.28 | 36 | 61 | 59[a] | 12 | 22 | 55 | 0 | 49 | 0 | – | – | – |
| 08.30 | 42 | 80 | 53 | 5 | 20 | 25 | 0 | 52 | 0 | – | – | – |
| 09.20 | 27 | 118 | 23[a] | 7 | 25 | 28 | 0 | 51 | 0 | – | – | – |
| 10.26 | 0 | 198 | 0 | NA | NA | NA | 16 | 52 | 31[a] | – | – | – |
| 12.08 | 0 | 89 | 0 | 0 | 54 | 0 | 0 | 53 | 0 | – | – | – |
| 2011.06.21 | 40 | 78 | 51 | 1 | 36 | 3 | 0 | 54 | 0 | 4 | 4 | 100[a] |
| 07.23 | 9 | 63 | 14 | 14 | 30 | 47 | 0 | 53 | 0 | 4 | 5 | 80 |
| 08.22 | 10 | 63 | 16 | 14 | 40 | 35 | 1 | 43 | 2 | 1 | 7 | 14 |
| 10.05 | 10 | 65 | 15 | 15 | 37 | 41 | 18 | 46 | 41[a] | 0 | 6 | 0 |
| 11.09 | 0 | 72 | 0 | 2 | 34 | 6 | 15 | 40 | 40[a] | 0 | 6 | 0 |
| 11.28 | 5 | 52 | 10 | 0 | 31 | 0 | 0 | 44 | 0 | NA | NA | NA |
| 12.16 | 0 | 63 | 0 | 0 | 36 | 0 | 0 | 40 | 0 | 0 | 9 | 0 |
| 2012.02.12 | 0 | 82 | 0 | 0 | 52 | 0 | 0 | 47 | 0 | 0 | 8 | 0 |

**Notes.**
[a] Indicates observation of developed ovaries in specimens.

*2006*; AB219220, AB219221) formed a monophyletic subclade with two novel sequences from this study (KX389366, KX389368; ML = 64%, NJ = 63%, $B = 1.0$).

*ITS-rDNA.* The phylogenetic tree resulting from maximum likelihood analyses of the ITS-rDNA sequence alignment is shown in Fig. 6C. *Palythoa tuberculosa and P.* sp. yoron formed a very well supported monophyletic clade (ML = 95%, NJ = 99%, $B = 0.96$). Within this clade were two comparatively well supported sub-clades, one made by sequences obtained only from *P.* sp. yoron sequences (=KX389470, KX389471, DQ997921; ML = 90%, NJ = 99%, $B = 1.0$), and the other including three *P. tuberculosa* sequences (DQ997909, DQ997929, DQ997919; ML = 70%, NJ = 83%, $B = 0.97$). *Palythoa mutuki* was paraphyletic and two well supported clades that included sequences from both *P. mutuki* and *P.* aff. *mutuki* were present (KX389473, KX389474, KX389475, KX389476, KX389481; ML = 93%, NJ =99%, $B = 1.0$; and DQ997892, KX389479, KX389480, KX389483; ML =72%, NJ = 77%, $B = 1.0$).

*ALG11.* The phylogenetic tree resulting from maximum likelihood analyses of the ALG11 sequence alignment is shown in Fig. 6D. Compared to the above phylogenetic trees, this tree was the most admixed, regardless of morphospecies. For example, sequences from *P.* sp. sakurajimensis (used as outgroup here) appeared throughout the tree. Only three terminal clades showed high bootstrap values (KX389373, KX389374, KX389379; ML =80%, NJ =86%, $B = 1.0$; and KX389403, KX389422; ML =90%, NJ =95%, $B = 1.0$; and KX389414, KX389418, KX389422; ML =78%, NJ =78%, $B = 1.0$).

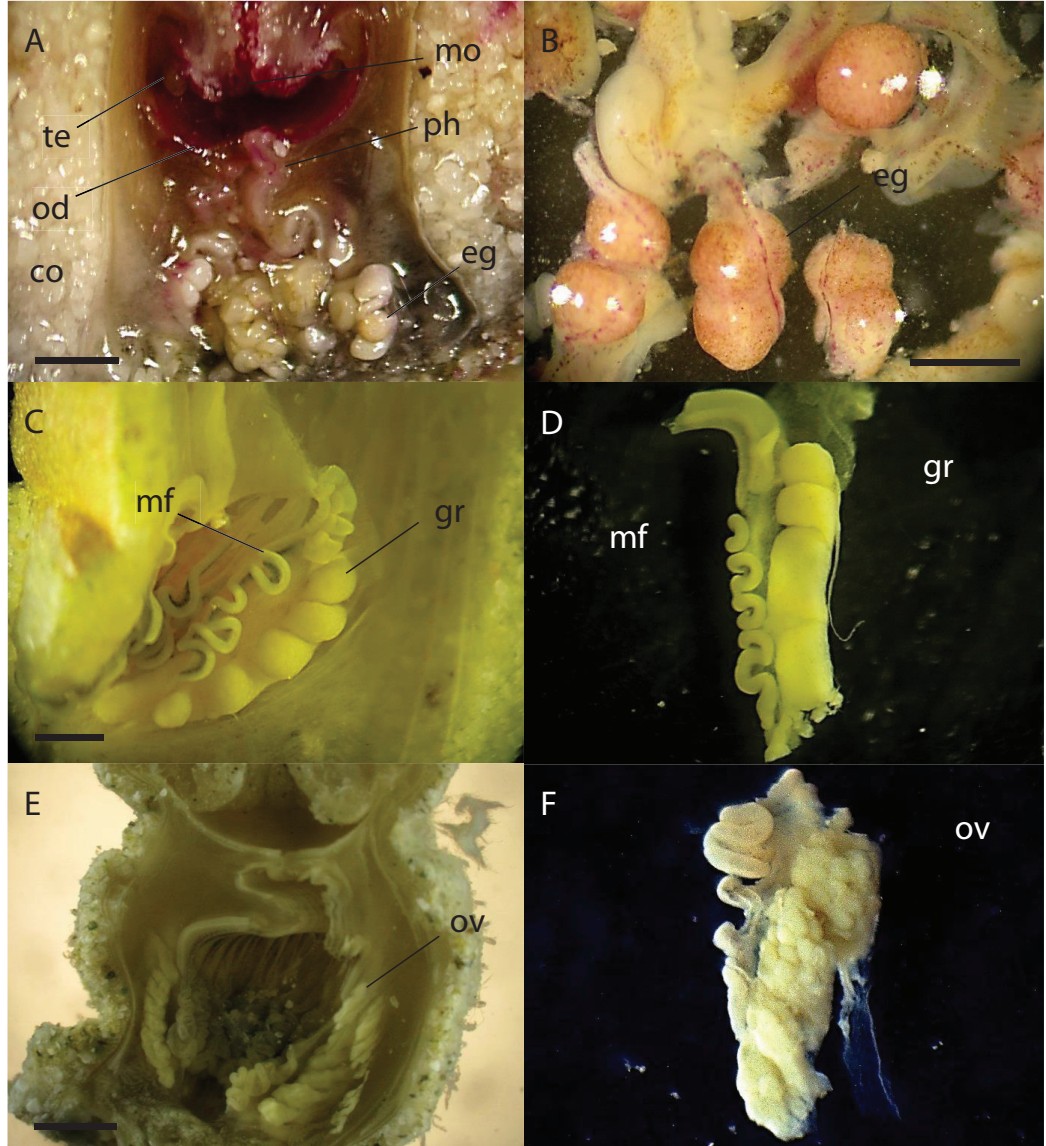

**Figure 5  Cross sections of *Palythoa* sp. *yoron* and *P.* aff. *mutuki* showing ovary development.** Cross section of polyp of (A) *Palythoa* sp. yoron (26 October 2010) and (B) matured eggs; (C) *P.* sp. yoron (9 November 2011) and (D) germinal ribbon inside a mesentery; (E) *P.* aff. *mutuki* (21 June 2011), and (F) developed ovaries. Abbreviations: te, tentacles; od, oral disk; co, coenenchyme; mo, mouth; ph, pharynx; eg, eggs; mf, mesenterial filament; gr, germinal ribbon; ov, ovary. Scale bars: 2 mm in (A) and (E) 500 μm in (B) 1 mm in (C, D and F) All images taken by M Mizuyama.

*Topology comparison between trees.*  Examining the two outgroups used in this study, *Palythoa* sp. sakurajimensis was phylogenetically much closer to *P. tuberculosa*, *P.* sp. yoron, *P. mutuki* and *P.* aff. *mutuki* compared to *P. heliodiscus* in every gene tree. There were few differences in sequences from the other four species groups, with only one base pair difference in the mtCOI tree, resulting in *P.* sp. sakurajimensis' sequences forming one group with some *P. mutuki* specimens, and only one to two base pairs' difference in

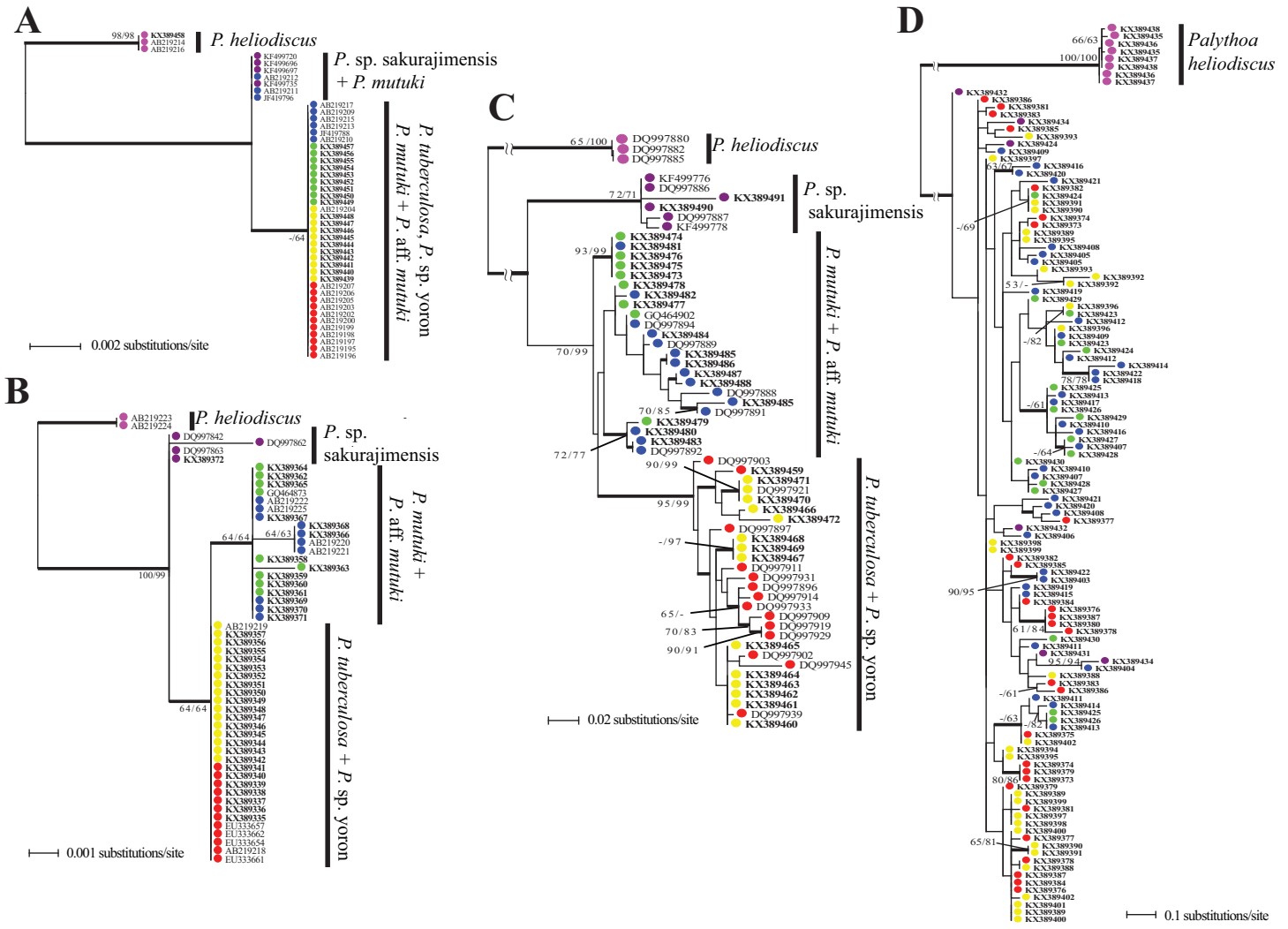

**Figure 6** **Phylogenetic trees of four DNA markers for *Palythoa* species examined in this study.** (A) Maximum likelihood (ML) tree of cytochrome oxidase subunit I (COI) sequences. (B) ML tree of mitochondrial 16S ribosomal DNA (mt16S rDNA) sequences. (C) Maximum likelihood tree of internal transcribed spacer of ribosomal DNA (ITS-rDNA) sequences. (D) Maximum likelihood tree of asparagine-linked glycosylation 11 protein (ALG11) region. Values at branches represent ML and NJ bootstrap probabilities, respectively (>50%). Bayesian posterior probabilities of >0.95 are represented by thick branches.

the mt 16S-rDNA tree for all four species groups. In particular, in the ALG11 tree, *P.* sp. sakurajimensis' sequences were admixed with the other four species groups.

*Palythoa tuberculosa* and *P.* sp. yoron (designated as the "*Palythoa tuberculosa* group" here), and *P. mutuki* and *P.* aff. *mutuki* (designated as "*Palythoa mutuki* group" here) did not separate into four species groups in each DNA marker's tree. The *P. tuberculosa* group formed a monophyletic clade in the ITS-rDNA tree and one grouping in the mt 16S-rDNA gene tree with one base difference from the *P. mutuki* group. On the other hand, the *P. mutuki* group did not show any common pattern, i.e., admixed with all other species groups except for *P. heliodiscus* in the ALG11 gene tree, most sequences forming

one monophyletic clade with the *P. tuberculosa* group due to no differences in sequences with some sequences forming one group with *P.* sp. sakurajimensis due to a one base pair difference from other specimens in the mtCOI tree, forming a monophyletic clade with one subclade in the mt 16S-rDNA tree, and forming a paraphyletic clade with a monophyletic subclade of *P. tuberculosa* in the ITS-rDNA tree.

*Species trees.* All hypothetical species were fully supported with posterior probability under both the four and six species models (Figs. 7A, 7B). The divergence time from the most recent common ancestor of *P. tuberculosa*, *P.* sp. yoron, *P. mutuki* and *P.* aff. *mutuki*, (divergence of *P.* sp. sakurajimensis in both cases), was calculated as 147,000 years before present with 95% credible interval [lower 30,900–upper 292,000] under the six species model and as 113,000 years under the four species model with 95% credible interval [lower 25,500–upper 231,000].

## DISCUSSION

The purpose of this study was to re-evaluate the systematics of some *Palythoa* species using an integrative approach. Primary hypotheses of species delimitation were based on external morphology (phenetic criterion) and habitat preferences (ecological criterion). These hypotheses were then examined in the light of additional characters, namely the number of tentacles, spawning periods and genetic data.

### Morphology and plasticity

The mean numbers of tentacles were significantly different among specimens of the four putative species; *P. tuberculosa*, *P.* sp. yoron, *P. mutuki* and *P.* aff. *mutuki* (Table 3). However, in previous research, the tentacle number of *P. tuberculosa* has been reported as various ranges, i.e., 30 to 40 (*Klunzinger, 1877*), up to 50 (*Walsh & Bowers, 1971*), 38 to 52 (*Reimer & Todd, 2009*), 30 to 37 (*Shiroma & Reimer, 2010*), or 30 to 50 (*Hibino et al., 2013*). A wider range of variations has been reported in *P. mutuki*, with 88 to 144 (*Ryland & Lancaster, 2003*), 60 to 74, approximately 80 for *P. mutuki*-related (*Reimer & Todd, 2009*), or 42 to 66 (*Shiroma & Reimer, 2010*) reported. Thus, the ranges of tentacle numbers can be assumed to be 30 to 52 for *P. tuberculosa* and 42 to 144 for *P. mutuki*, and therefore tentacle numbers of *P.* sp. yoron and *P.* aff. *mutuki* observed in this study are within ranges of previously reported intraspecific variation. These differences between tentacle numbers reported in the literature and our data may be partly explained by the fact that previous authors did not consider *P.* sp. yoron and *P.* aff. *mutuki* as different species.

However, *Ong, Reimer & Todd (2013)* also demonstrated phenotypic plasticity in *P. tuberculosa* with high ability to acclimate against changes in light-induced environments. From *in situ* observations, *P.* sp. yoron seems to prefer locations exposed to strong current such as extensive reef flats where the back reef moat is widely developed. Correspondingly, *P.* sp. yoron is also often found in back reef moats, as *Shiroma & Reimer (2010)* mentioned, covered with sand or other loose detritus. High numbers of tentacles enable them to acquire nutritious detritus and feed on planktonic organisms, but strong-current environments repeatedly cover colonies with sand. From the viewpoint of its small, tetrapod colony shape,

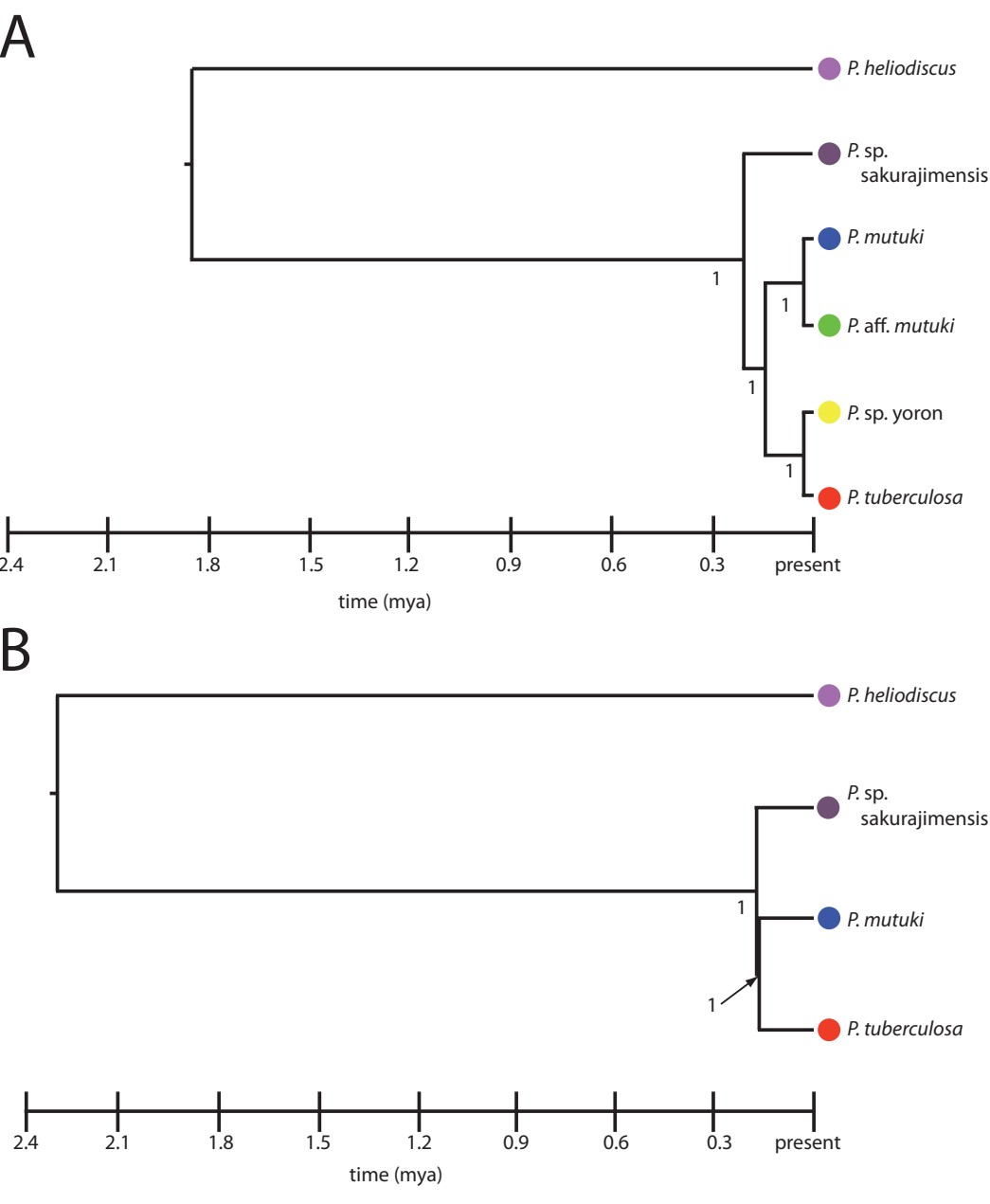

**Figure 7** **Species trees for *Palythoa* under (A) six species model, and (B) four species model.** Values at branches represent posterior probability.

*P.* sp. yoron seems have adapted to such an environment. Therefore, to ensure whether differences in tentacle numbers and colony form between *P. tuberculosa* and *P.* sp. yoron are caused by species differentiation, the observation of reaction norms of each species with transplantation experiments is needed.

Although previous cnidae research with detailed statistical analyses revealed finer-scale differences among *Palythoa* species (*Ryland & Lancaster, 2004*), we did not observe any useful diagnostic differences with utility for rapid identification of species groups in this study.

## Spawning periods and reproductive isolation

Over the two years analyzed, *P.* sp. yoron consistently developed ovaries later than the three other putative species. If we assume a sharp drop in the proportion of developed ovaries as the consequence of the release of eggs, the annual spawning period estimated for *P.* sp. yoron was early to mid-November and that of *P.* aff. *mutuki* mid- to late June. The spawning period of *P. tuberculosa* in Okinawa-jima I. has been reported in early August (*Yamazato, Yoshimoto & Yoshihara, 1973*), from the end of July to middle August (*Shiroma & Reimer, 2010*), and on 19 and 20 August in 2009 (*Hirose et al., 2011*). In our study, spawning was estimated to have occurred in August in 2010 and possibly from early July in 2011. The reproductive season of *P. mutuki* was presumed that be synchronized with *P. tuberculosa* in 2010, although developed eggs were not confirmed. Little is known about the sexual reproductive ability of this species, and according to *Ryland & Lancaster (2003)* the only previous records of *P. mutuki* possessing developed oocytes are from Fiji and Tuvalu. To overcome this lack of knowledge, closer examinations via staging of histological sections for gonadal development (such as done by *Polak et al., 2011*) are required.

Interpreting these results in terms of putative reproductive isolation is not straightforward. Even assuming that a sharp drop in the proportion of developed ovaries translates into a major spawning event, which seems to be a reasonable hypothesis, this does not exclude the possibility of eggs being released much later than the initial peak. For example, while we estimated the spawning period of *P. tuberculosa* to have occurred in August in 2010, nearly 20% of individuals still had developing or developed ovaries on September 20th, which may have been released as mature eggs at any time from then until October 26th (Fig. 4A), and enabled potential cross-fertilization with *P.* sp. yoron. On the other hand, data thus far indicate spawning on one or two nights per year for brachycneminic zoantharians (*Ryland, 1997*), and reabsorption of oocytes (Ono et al. 2005) that did not spawn. More work is needed to determine exact spawning patterns of *Palythoa tuberculosa* and closely *Palythoa* related species, but the asynchrony of both ovary development (*P.* sp. yoron) and spawning peaks for *P. tuberculosa* and *P.* aff. *mutuki* suggest that at least partial pre-zygotic reproductive isolation is possible among *P.* sp. yoron, *P. tuberculosa* and *P.* aff. *mutuki* at Tokunoshima I.

## Species boundaries in phylogenetic trees

The four genetic markers analyzed in this study displayed contrasting patterns. The two mitochondrial genes were relatively conservative, as has been reported for other anthozoans (*Shearer et al., 2002*; *Huang et al., 2008*), but mt 16S-rDNA allowed the recovery of *P. heliodiscus*, *P.* sp. sakurajimensis, the *P. mutuki* group and the *P. tuberculosa* group as four genetically homogeneous groups (phenetic criterion), and all species or species groups were reciprocally monophyletic with the exception of *P. tuberculosa*. ITS-rDNA showed a similar pattern with the *P. mutuki* group and the *P. tuberculosa* group represented in

distinct clades, although the *P. mutuki* group was paraphyletic. This consistency across mitochondrial and nuclear markers also suggests that there is no genetic exchange (biologic criterion) between these four groups, and thus provides a first level of species delimitation. In contrast, all *Palythoa* spp. besides *P. heliodiscus* were largely mixed in the tree recovered from the ALG11 marker, which strongly suggests incomplete lineage sorting for this gene.

Despite obvious differences in morphology and reproductive season between *P. tuberculosa* and *P.* sp. yoron, as well between *P. mutuki* and *P.* aff. *mutuki,* no molecular marker was successful in dividing these species pairs into their own monophyletic clades. *Palythoa* sp. yoron formed a subclade from two specimens in *Reimer et al. (2007a)*, however, in this study reconstructing phylogenetic trees based on the same genomic region with more specimens of *P.* sp. yoron, one mixed monophyletic clade was supported well with all the other *P. tuberculosa* specimens. The same pattern was observed with *P. mutuki* and *P.* aff. *mutuki*. These results imply either gene flow between each pair of nominal species or incomplete lineage sorting. Although these two alternative hypotheses are not mutually exclusive, the absence of intermediate morphotypes and the presence of distinct spawning periods lead us to favor the latter over extensive gene flow. Sequences from other single-copy nuclear markers like ALG11 are required to more thoroughly resolve these two species pairs.

## Sympatric speciation timing

Recently, sympatric speciation has come to be understood as a major generator of marine biodiversity (reviewed in *Bowen et al., 2013*). Under such situations, ecological (e.g., behavior or microhabitat) boundaries lead to isolation. However, the hierarchy of timing of sympatric speciation processes (e.g., the order that separation occurs via phylogenetic, reproductive, and morphological criteria) as lineages diverge remains not well understood, with no clear consensus (*Norris & Hull, 2012*; *Pabijan et al., 2017*). For example, in tropical bivalves, phylogenetic differences (=cryptic species) have been observed without any clear evidence of morphological differences (e.g., *Lemer et al., 2014*). On the other hand, in many marine taxa, it has been proposed that during sympatric speciation, reproductive isolation is one driving force behind lineage divergence (*Palumbi, 1994*).

In this study, morphology and reproductive data sets showed four *Palythoa* lineages, while DNA markers showed either two lineages (ITS-rDNA, mtCOI, mt 16S-rDNA) or one admixed lineage (ALG11). Combined molecular analyses suggested either two or four lineages were equally possible (Fig. 7). Such varied results along a speciation continuum between different datasets reflect the patterns to be expected during ongoing or incomplete speciation events (*Nosil, Harmon & Seehausen, 2009*). As all four *Palythoa* lineages can be found in sympatry at Tokushima I., our results suggest that reproductive isolation, perhaps caused by past hybridization and back-crossing events (*Reimer et al., 2007a*; *MacLeod et al., 2015*), led to the generation of these different lineages and morphological differentiation. Phylogenetic differentiation currently remains incomplete due to the evolutionary recentness of these events, estimated as less than 200,000 years before present. Such confounding data, with reproductive isolation but incomplete genetic lineage sorting, can be expected due to the extended duration of speciation events (*Norris & Hull, 2012*).

## CONCLUSIONS

Overall, the data imply that *Palythoa* species have a much more complex evolutionary history at the species level than previously expected (e.g., in *Reimer et al., 2007a*). However, natural hybridization between *P. tuberculosa, P.* sp. yoron and *P.* aff. *mutuki* seems to not be currently occurring, at least for populations at Tokunoshima I. observed in this study. In spite of ambiguous phylogenetic differentiation between *P. tuberculosa* and *P.* sp. yoron, and between *P. mutuki* and *P.* aff. *mutuki*, we consider these four lineages are all distinct species based on their morphological differentiation and distinct spawning periods. *In situ* observation of spawning events combined with genomic level examinations will help further clarify the hierarchy of timing in speciation events, and these four sympatric *Palythoa* lineages present a potential model system for such studies.

## ACKNOWLEDGEMENTS

The authors sincerely thank Dr. D Albinsky (University of the Ryukyus, UR) for technical help during molecular experiments. The people of Tokunoshima I. are thanked for help during field surveys. Also, thanks to Dr. M Obuchi and Dr. A Iguchi (both UR), who both spared much time for discussion of statistical analyses. Dr. M Maronna (U. Sao Paolo) is thanked for comments on alignments. Finally, MISE Laboratory members are thanked for their support. Comments from three reviewers and the editor greatly improved an earlier version of this manuscript.

### Funding

The senior author was funded by the International Research Hub Project for Climate Change and Coral Reef/Island Dynamics at UR, by a Japan Society for the Promotion of Science (JSPS) 'Zuno-Junkan' grant entitled 'Studies on origin and maintenance of marine biodiversity and systematic conservation planning', and by a JSPS Kiban B grant entitled 'Global evolution of Brachycnemina and their *Symbiodinium*'. The funders had no role in study design, data collection and analysis, decision to publish, or preparation of the manuscript.

### Grant Disclosures

The following grant information was disclosed by the authors:
International Research Hub Project.
'Zuno-Junkan' grant.
JSPS Kiban B grant.

### Competing Interests

James D. Reimer is an Academic Editor for PeerJ.

## Author Contributions

- Masaru Mizuyama conceived and designed the experiments, performed the experiments, analyzed the data, prepared figures and/or tables, authored or reviewed drafts of the paper, approved the final draft.
- Giovanni D. Masucci performed the experiments, analyzed the data, prepared figures and/or tables, authored or reviewed drafts of the paper, approved the final draft.
- James D. Reimer conceived and designed the experiments, performed the experiments, analyzed the data, contributed reagents/materials/analysis tools, prepared figures and/or tables, authored or reviewed drafts of the paper, approved the final draft.

## DNA Deposition

The following information was supplied regarding the deposition of DNA sequences:

Novel sequences generated in this study are accessible via GenBank accession numbers KX389335–KX389491.

## Data Availability

The raw data are provided in the Tables.

## Supplemental Information

Supplemental information for this article can be found online at http://dx.doi.org/10.7717/peerj.5132#supplemental-information.

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
