# Peer review of "Speciation among sympatric lineages in the genus Palythoa (Cnidaria: Anthozoa: Zoantharia) revealed by morphological comparison, phylogenetic analyses and investigation of spawning period"

_PeerJ, doi:10.7717/peerj.5132_

## Round 0.1 · original submission · Minor Revisions

We now have 3 reviews back on your submission, and while generally positive, each have suggestions for improvement of the manuscript. These range from relatively minor revisions with suggestions for future directions to the criticism that environment is not properly controlled in this analysis and that the morphological differences may in fact be due to differences in microhabitat in which the individuals were collected. I do not know if environmental data are available for each of the colonies, but it seems to me that adding such information if possible would address the concerns of the most critical referee and avoid future readers having the same response. This referee is expert in another group of cnidarians and has no vested interest in zoanthids, so the question of whether the genetic data support recognition of P. tuberculosa and P. sp. yoron is likely to be raised by others also. Given that, it seems to me that being able to address the comments of this referee will go a long way towards convincing objective readers from outside this field that the taxonomy reported here is correct. I expect that the revisions requested are largely semantic and do not require additional data or extensive reanalyses of the existing data, so I consider this a relatively minor revision.

I look forward to seeing the revised manuscript.

Reviewer 1 ·

Basic reporting

This study is deserve to be published in Peer J after minor revision.

Experimental design

Experimental design is good enough to be published. However, I suggest the authors to add extra experiments and analyses in the future study if authors hope to do them.

Validity of the findings

Finding is deserve to be published after minor revision.

Additional comments

This study, entitled “Speciation among sympatric lineages in the genus Palythoa (Cnidaria: Anthozoa: Zoantharia) revealed by morphological comparison, phylogenetic analyses and investigation of spawning period”, attempted to delineate Palythoa species boundaries and to clarify the relationship among four species groups by multiple independent criteria. The manuscript is clearly written and unambiguous language. And the data presented in this study are enough to indicate the complexity of speciation with hybridization, and I believe the results by this study are very informative for speciation study not only about Cnidaria but also other organisms. In my opinion, I agree with author’s conclusion, the sentence in abstract as follows; “the morphological and reproductive results suggest these lineages are four separate species, and that incomplete genetic lineage sorting may prevent the accurate phylogenetic detection of distinct species with the DNA markers utilized in this study, demonstrating the value of morphological and reproductive data when examining closely related lineages”.
I have two suggestions for your future study; first, you may try to observe “real” spawning about your targeted four species group to confirm the extent of pre-zygotic reproductive isolation. There are some papers about sexual reproduction of Palytoa tuberculosa including spawning time (i.e. Hirose et al. 2011; Polak et al. 2011), so it may not be difficult for you to try it. And if possible, you may try heterospecific crossing to clear whether stronger pre-zygotic reproductive isolation is occurred or not.
Second, you may try haplotype network analysis using exon region and structure analysis using neutral region such as MS and/or SNP. Of course, these analyses will not be able to distinguish between hybridization on this time and incomplete lineate sorting, however, you may take outline of some extent of genetic introgression among species groups.


Minor points

Line 77: tuberculosa and P. mutuki. (Fig.1, Table 1).
Change “Fig. 1” to “Fig. 2”.

Line 122-135
Did you check ovary only? You should indicate the reason why that you did not check testis in addition to that these Palythoa are hermaphrodite or gonochoric.

·

Basic reporting

The manuscript entitled “Speciation among sympatric lineages in the genus Palythoa (Cnidaria: Anthozoa: Zoantharia) revealed by morphological comparison, phylogenetic analyses and investigation of spawning period” submitted to PeerJ was evaluated. This manuscript is quite dense with several very relevant approaches. The authors attempt to reconstruct, based on molecular data and patterns of gonadal development, the evolutionary history of the species recognized for the study area. The manuscript in general is very good and the authors were very careful in the elaboration of the materials and methods, something increasingly rare these days. Anyway, I have some issues that should be addressed before the final acceptance of the manuscript.

One point that bothered me a bit was the text that justified the non-inclusion of other ALG11 sequences. The authors state “…except for ALG11, as no closely related sequences (e.g. Anthozoa) were included in the dataset”. This statement is not correct, as there are a few dozen anthozoan sequences of ALG11 on Genbank, including Zoantharia sequences. In this way, I suggest to the authors the reformulation of this definition (I do not think it is useful to include sequences in this analysis), since it passes incorrect information.

Another point that I think deserves attention is the phylogenetic trees presented. I think that deserve to be in isolated figures. The authors presented the reconstructions in separate topics for each marker, so I think that trees must be independent figures, because that can help in reading the text. I assume I was quite confused with the figure many times to be able to recognize each pattern.

Still on the figures, I suggest for authors to add the time scale in figure 7 (Species Tree). This is quite simple and can greatly help for future studies in biogeography.

Experimental design

The use of unusual molecular markers can be a very difficult factor. However, the authors addressed this issue very well. Anyway, I do not totally agree with the adoption of filtering methods in a very short sequences dataset. In this way, I suggest to the authors the inclusion (maybe as supplementary material) of an alignment via MAFTT (without any filtering) and new phylogenetic reconstructions (perhaps only ML). In a reconstruction at the genomic level (transcriptomics, etc.) the use of filtering is acceptable (even recommended), but in very short sequences the loss of information may be too high. The inclusion of these data may help to verify variations caused by the phylogenetic reconstruction method and may also be useful for some future studies on the group.

Validity of the findings

no comment

Additional comments

After these modifications, fairly simple, the manuscript can be accepted for publication. The manuscript brings together very relevant data for publication and the discussion has enormous value not only for the group, but for understanding some evolutionary patterns in reef environments.

Reviewer 3 ·

Basic reporting

This manuscript is generally written in a clear manner. Occasionally the authors use ambiguous sentence structure and terms, which are noted below. References are sufficient for the reader to understand how the work fits into the broader field. The structure of the article is clear and easy to follow. The figures are of good quality overall, however, additional clarity is needed in figure 2. It would benefit the reader to clearly label each species within the figure to highlight morphological differences between the species. The raw sequence data is freely available. However, it is unclear when reading the methods section that additional data from genbank was used in this study, this should be more apparent to the reader. The description on lines 185-187 obscures this, as the number of sequences generated in this study is reported and then they are aligned with sequences in genbank. It would be much easier to follow if that paragraph ended with the total combined number of sequences for each locus.

Experimental design

The authors of this study are attempting to define a set of morphological and molecular markers that discern between closely related lineages in the genus Palythoa. The authors first collected individuals belonging to each lineage based on their morphological features and the local habitat. The authors then attempted to differentiate individuals from the four groups based on morphology (tentacle number, cnidae, and timing of reproduction) and using molecular tools including a newly developed nuclear marker (ALG11). For the most part, the methods are reproducible, although, as I describe below, could have gone further to address the questions at hand.

Validity of the findings

The conclusions of this paper based on the data the authors collected could have gone further to disentangle whether these lineages are species or populations.
1. These species are highly morphologically variable (lines 365-371) and the authors suggest that this contributes to difficulty in classifying species (lines 58-59). Furthermore, these species all occupy different “microenvironments” but are also found sympatrically (lines 73-77). The analysis would greatly benefit from careful statistical control to ensure that environmental variation is not contributing to statistically significant signals of phenotypic differences between these species. Especially given the fact that the molecular data (Fig. 6) do not overwhelmingly support differentiating these 4 lineages into species, it is crucial to control for environmental variation in the morphological differences the authors describe.
2. The molecular dataset does not seem to support the 4 lineages and the discussion should address this issue. First, the mitochondrial genes appear to be inadequate to address species-level differences. Slow rates of mitochondrial evolution are well known in cnidarians (e.g. Daly et al. 2008), which means that these markers are likely inadequate in discerning the relationships of recently diverged species. Second, if P. sp. yoron actually comes from a hybridization event, one would not expect it to form a monophyletic clade with P. tuberculosa in the ITS gene tree. Third, it is true that the ALG11 gene tree supports incomplete lineage sorting as well as hybridization between these groups. However, it seems difficult to rule out the possibility that the ALG11 gene tree reflects the fact that these lineages are all the same species. The most obvious signal in the molecular data appears to be that there are two groups (P. mutuki and P. aff. mutuki; and P. tuberculosa and P. sp. yoron). The discussion should address if these lineages should actually be divided, in light of the molecular data and taking into consideration that patterns in the morphological data could be due to environmentally driven phenotypic plasticity.
3. It is concerning that some loci are entirely missing from some species, including the ITS locus which appears to be more informative about relationships than the mitochondrial loci. Although the authors supplemented the dataset with sequences from genbank, the manuscript would be greatly improved if the individuals that were collected for the purposes of this study were the source of the genetic data. This is especially crucial given that the study is aiming to find morphological and molecular differences between these lineages. The inference would be strengthened, for example, if the authors could definitively state that the individuals that they used in the morphological portion of the study fell within the clades shown in Figure 6.
4. The statistical analysis of the morphological differences between the species would be improved by using a model that accounted for the replication within and between colonies. The sample size appears to be high in the table reporting these differences but they come from sampling 7-13 colonies of each species. The authors should consider using a statistical test that accounts for the replication within colonies when testing for differences between the species.
5. The data that the authors present here could provide evidence for whether Palythoa sp. yoron is the result of a hybrid speciation event. The authors cite previous work (Reimer et al. 2007a) that this species appears to have increased levels of variation (“shared additive patterns”) due to the hybridization event. This hypothesis could be further tested with the addition of another nuclear marker (ALG11). However, the authors should be cautious if they use this pattern to argue for hybrid speciation. As the authors point out, shared variation can also arise from incomplete lineage sorting, and increased nucleotide diversity could arise from the populations experiencing different demographic histories. It will be difficult to differentiate between these alternatives with so few loci.
6. The estimated divergence time is reported as 147,000 ybp or 113,000 ybp depending on the model. Uncertainty estimates around these numbers would make these easier to interpret for the reader and would strengthen the argument that divergence was relatively recent.

Additional comments

Line 68: Consider removing “apparently”
Line 77: Do you mean to refer to Fig. 2 instead of 1?
Line 107-108: The final sentence of the paragraph is redundant, please consider removing it.
Ling 123: Remove “in”
Line 134-135: Final sentence is awkward, maybe consider changing to “The end of the estimated spawning period was defined as the point where the number of developed/developing ovaries reached 0%.”
Line 148-152: This sentence is confusing, why is the marker better for resolving relationships if it has no introns?
Line 203: change “codons” to “codon positions”
Line 204: The phrase “because of homogenizing substitution rates between loci” is confusing, did you do this to avoid homogenizing substitution rates because they are known to vary by codon position?
Line 268-271: This sentence is confusing. It is difficult to determine when small holotrichs were present and absent based on the construction of the sentence.
Line 422: “three first” is awkward, is it the three first species? It might be clearer to write: “all species or species groups were reciprocally monophyletic with the exception of P. tuberculosa”
Line 437: Should “within” be “between?”
Line 445: The phrase “hierarchy of timing” is used multiple times (e.g. Lines 465-466) and it is unclear what that means, please consider adding a brief explanation the first time you use it to guide the reader.
Line 450: This statement is likely not only true in marine lineages, reproductive isolation can be a driving force in sympatric speciation in terrestrial lineages too. Please consider adding a reference to reflect this.
Figure 6 caption: Please consider revising “highest posterior density interval,” explain what the interval means.

---

## Round 0.2 · Minor Revisions

Dear Masaru and co-authors,

I have now received one last assessment of your work from reviewer 3 and have also reviewed the manuscript myself. Overall, this manuscript is of good quality and the work will constitute an excellent contribution to the field. I have left a number of minor edits/comments for your consideration in the attached version (Editor Annotated), please incorporate them as you see fit.

Figure 6 needs improvement. This is an extremely busy Figure and it is currently impossible to identify the sequence names without getting a headache (see my comments in Figure 6). I suggest increasing the fonts (where possible), and either indicate sequence accession numbers in place of sequence names, or create sequence numbers (e.g. #1, #2, #3, etc..) which could then be linked back to the Table(s) where these specific sequences are detailed.

Once you have addressed the above concerns, I will be happy to accept this manuscript for publication in PeerJ.

With kind regards,
Xavier

Reviewer 3 ·

Basic reporting

Basic reporting is fine, the phrasing and paragraph structure that was previously unclear is much improved. However, there was one sentence that is unclear on lines 340-346. The purpose appears to be to show how and why the P. mutuki group is not monophyletic but it is very difficult to parse. Although the paragraph would be less compact, it would be easier to read if that sentence was broken up into smaller sentences.

Experimental design

The experimental design is within the aims and scope of the journal. The questions are well defined and the approach to addressing them is clear throughout the paper.

Validity of the findings

The findings are valid and the statistical analyses are sound. The conclusions are well-stated and are reasonable given the patterns in the data.

---

## Round 0.3 · accepted · Accept

Dear Masaru, Giovanni, and Jamie,

Thank you very much for your revised manuscript. I am satisfied with your modifications and am delighted to accept this work for publication in PeerJ. Thank you for your hard work and nice contribution to the field, well done!

With warm regards,
Xavier